# Once-for-All: Controllable Generative Image Compression with Dynamic Granularity Adaptation

**Anqi Li** [1,2] **Feng Li** [3,*] **Yuxi Liu** [1,2] **Runmin Cong** [4] **Yao Zhao** [1,2] **Huihui Bai** [1,2,*]

[1] Institute of Information Science, Beijing Jiaotong University
[2] Beijing Key Laboratory of Advanced Information Science and Network Technology
[3] School of Computer Science and Engineering, Hefei University of Technology
[4] School of Control Science and Engineering, Shandong University
`{lianqi, yuxiliu, yzhao, hhbai}@bjtu.edu.cn`
`fengli@hfut.edu.cn, rmcong@sdu.edu.cn`

## Abstract

Although recent generative image compression methods have demonstrated impressive potential in optimizing the rate-distortion-perception trade-off, they still face the critical challenge of flexible rate adaptation to diverse compression necessities and scenarios. To overcome this challenge, this paper proposes a **Control**lable **G**enerative **I**mage **C**ompression framework, termed **Control-GIC**, the first capable of fine-grained bitrate adaptation across a broad spectrum while ensuring high-fidelity and generality compression. Control-GIC is grounded in a VQGAN framework that encodes an image as a sequence of variable-length codes (*i.e.* VQ-indices), which can be losslessly compressed and exhibits a direct positive correlation with bitrates. Drawing inspiration from the classical coding principle, we correlate the information density of local image patches with their granular representations. Hence, we can flexibly determine a proper allocation of granularity for the patches to achieve dynamic adjustment for VQ-indices, resulting in desirable compression rates. We further develop a probabilistic conditional decoder capable of retrieving historic encoded multi-granularity representations according to transmitted codes, and then reconstruct hierarchical granular features in the formalization of conditional probability, enabling more informative aggregation to improve reconstruction realism. Our experiments show that Control-GIC allows highly flexible and controllable bitrate adaptation where the results demonstrate its superior performance over recent state-of-the-art methods.

## 1 Introduction

Lossy image compression complies with the rate-distortion criterion in Shannon's theorem (Shannon et al., 1959), which aims to pursue minimal storage of images without quality sacrifice. Traditional standardized codecs (Wallace, 1990; Taubman & Marcellin, 2001; Bellard) adhere to a typical hand-crafted "transforming-quantization-entropy coding" rule, showing substantial performance on generic images. Learnable compression algorithms (Ballé et al., 2017; 2018; Minnen et al., 2018; Minnen & Singh, 2020) follow a similar pipeline that parameterizes it as convolutional neural networks (CNNs) operating on latent variables with end-to-end R-D optimization. Recent works (Santurkar et al., 2018; Tschannen et al., 2018; Agustsson et al., 2019; Mentzer et al., 2020) leverage generative adversarial networks (GANs) (Goodfellow et al., 2014) to deal with the compression task, known as generative image compression, which minimizes the distribution divergence between original and reconstructed images, producing perfect perceptual quality. However, these methods train the models separately for specific R-D points with Lagrange multiplier ($\lambda$)-weighted R-D loss, each corresponding to an individual $\lambda$. In this way, multiple fixed-rate models are necessitated to vary bitrates, leading to

---

* Corresponding Authors

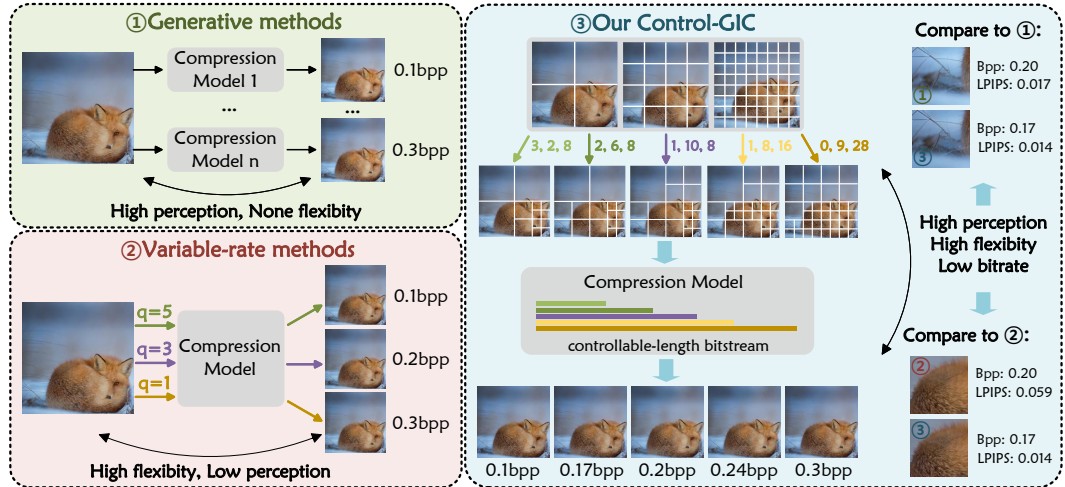

Figure 1: Illustration of the key motivation behind our approach. ① Generative methods train separate models for each distinct compression ratio, which achieves promising perceptual quality but overlooks the flexibility. ② Variable-rate methods modify the compression model by introducing truncated quantization parameters, which only support a limited range of bitrates and cannot balance the perceptual quality and compression efficiency. ③ Our proposed Control-GIC enables the generation of a controllable-length bitstream following different granularity decisions of image patches, which achieves an excellent trade-off among flexibility, perceptual quality, and compression efficiency.

dramatic computational costs and inefficient deployment to cater to diverse bitrates and devices. Some CNN-based models propose to learn scalable bitstreams (Johnston et al., 2018; Toderici et al., 2017; Bai et al., 2021; Mei et al., 2022; Zhang et al., 2024a; Jeon et al., 2023b) or truncated quantization parameters to control the bitrates (Toderici et al., 2016; Choi et al., 2019; Yang et al., 2021; Cui et al., 2021). On the one hand, these models typically support a limited range of bitrates with substantial variance distribution, thus constraining their adaptability to finer bitrate adjustments. On the other hand, they mostly quantify the distortion using mean square error (MSE), which is inconsistent with human perception and often yields implausible reconstruction, particularly at low bitrates (Blau & Michaeli, 2019; Mentzer et al., 2020). Several methods introduce scalable (Iwai et al., 2024) or variable-rate (Guo et al., 2023) designs into generative models. While achieving remarkable improvements in perceptual quality, they are still constrained by finite compression rates (see Figure 1).

In light of the preceding discussion, this work proposes an innovative generative image compression paradigm, dubbed **Control-GIC**, which accommodates highly flexible and fine-grained controls on a broad range of bitrates and perceptually realistic reconstruction with solely one set of optimized weights. Motivated by that VQ-based models (Esser et al., 2021; Zheng et al., 2022) enable to encode images into discrete codes representing the local visual patterns, Control-GIC hybridizes the classical coding principle in the architecture with VQGAN to relax the typical R-D optimization and provide a controllable unified generative model. Specifically, Control-GIC first characterizes the inherent information density and context complexity of local image patches as the information entropy. We devise the granularity-informed encoder that determines the granularity of these patches based on their entropy values. These are further represented by sequential variable-length codes (*i.e.*, VQ-indices) based on learned codebook prior. One can flexibly control the statistics of the granularities to adjust the VQ-indices of patches dynamically adapting to diverse desirable bitrates. As correlated to the regional information of images, the VQ-indices are spatially variant to adapt to the local contents. We then develop a no-parametric statistical entropy coding module, which captures the code distribution in the codebook prior across a large-scale natural dataset to approximate a generalized probability distribution. This enables lossless and more compact encoding of VQ-indices during inference, improving the compression efficiency. On top of entropy coding, a probabilistic conditional decoder is further developed, which formalizes the reconstruction of granular features in a conditional probability manner with historic encoded multi-granularity representations given entropy-decoded indices. Our comprehensive experimental results demonstrate the outstanding

adaptation capability of Control-GIC, which achieves superior performance from perceptual quality, flexibility, and compression efficiency over three types of recent state-of-the-art methods including generative, progressive, and variable-rate compression methods using only a single unified model.

The main contributions of this work are three-fold:

- We propose **Control-GIC**, a unified generative compression model capable of variable bitrate adaptation across a broad spectrum while preserving high-perceptual fidelity reconstruction. To our knowledge, this is the first that allows highly flexible and controllable bitrate adaptation.

- We propose a granularity-inform encoder that represents the image patches of sequential spatially variant VQ-indices to support precise variable rate control and adaptation. Besides, a non-parametric statistical entropy coding is devised to encode the VQ-indices losslessly.

- We design a probabilistic conditional decoder, which aggregates historic encoded multi-granularity representations to reconstruct hierarchical granular features in a conditional probability manner, achieving realism improvements.

## 2 RELATED WORK

**Neural Image Compression**  Transformation, quantization, and entropy coding are three key components in neural image compression (NIC). Since Ballé *et al.* (Ballé et al., 2017) propose the pioneering learnable NIC method using convolutional neural network (CNN), later methods make improvements in transformation to learn a more compact and exact representation with efficient architecture designs (Cheng et al., 2020; Xie et al., 2021; Zou et al., 2022). Some researchers are dedicated to improving the entropy coding by designing hyperprior and context models (Ballé et al., 2018; Lee et al., 2018; Qian et al., 2022) with entropy model, which can capture more precise spatial dependencies in the latent, helping probability distribution estimation. In recent years, the integration of generative models like GANs (Goodfellow et al., 2014; Wang et al., 2018; Karras et al., 2019) and Diffusion Model (DM) (Ho et al., 2020; Zhang et al., 2024b; Wu et al., 2024) into NIC has shown promising results. For instance, Agustsson *et al.* (Agustsson et al., 2019) uses GAN loss along with R-D loss to achieve end-to-end full-resolution image compression while giving dramatic bitrate savings. Mentzer *et al.* (Mentzer et al., 2020) incorporates GAN with compression architecture systematically and generates robust perceptual evaluation. Yang *et al.* (Yang & Mandt, 2024) propose an end-to-end DM-based compression framework and reconstruct images through the reverse diffusion process conditioned with context information, outperforming some GAN-based methods. VQGAN (Esser et al., 2021)-based techniques (Mao et al., 2023; Xue et al., 2024; Jia et al., 2024) have demonstrated strong codebook priors for discrete visual feature representation in image synthesis, offering new insights for compression. Mao *et al.* (Mao et al., 2023) introduce VQ-indices compression for simple yet efficient compression, markedly improving the compression ratio. Building on these findings, we aim to harness the potential of VQ and customize VQGAN designs for controllable generative compression across various bitrates with a unified model.

**Rate-Adaptation NIC**  The aforementioned methods often face the challenge of deployment in resource-limited devices they are trained as separate models for specific bitrates, which increases the complexity overhead to support multiple bitrates. Current research solving such a rate adaptation problem can be roughly divided into two categories: variable-rate compression (Chen & Ma, 2020; Lu et al., 2021; Cui et al., 2021; Guo-Hua et al., 2023) and progressive compression (Toderici et al., 2017; Mei et al., 2022; Lee et al., 2022a; Jeon et al., 2023b; Zhang et al., 2024a). Variable-rate methods, such as those by Theis *et al.* (Toderici et al., 2017) and Choi *et al.* (Choi et al., 2019), adjust scalar parameters or use conditional convolutions to adapt to different quality levels. Others, like Cai *et al.* (Cai et al., 2019) and Yang *et al.* (Yang et al., 2021), employ multi-scale representations or slimmable networks for content-adaptive rate allocation. Cui *et al.* (Cui et al., 2021) and Lee *et al.* (Lee et al., 2022b) introduce gain units and selective compression, respectively, to further refine bitrate control. Progressive compression(Toderici et al., 2017; Johnston et al., 2018) develops scalable bitstreams for bitrate flexibility. Zhang *et al.* (Zhang et al., 2024a) propose to explore the receptive field with uncertainty guidance for both quality and bitrate scalable compression. Lee *et al.* (Lee et al., 2022a) propose to encode the latent representations into a compressed bitstream trip-plane to support fine-granular progressive compression. Jeon *et al.* (Jeon et al., 2023b) further improve it with

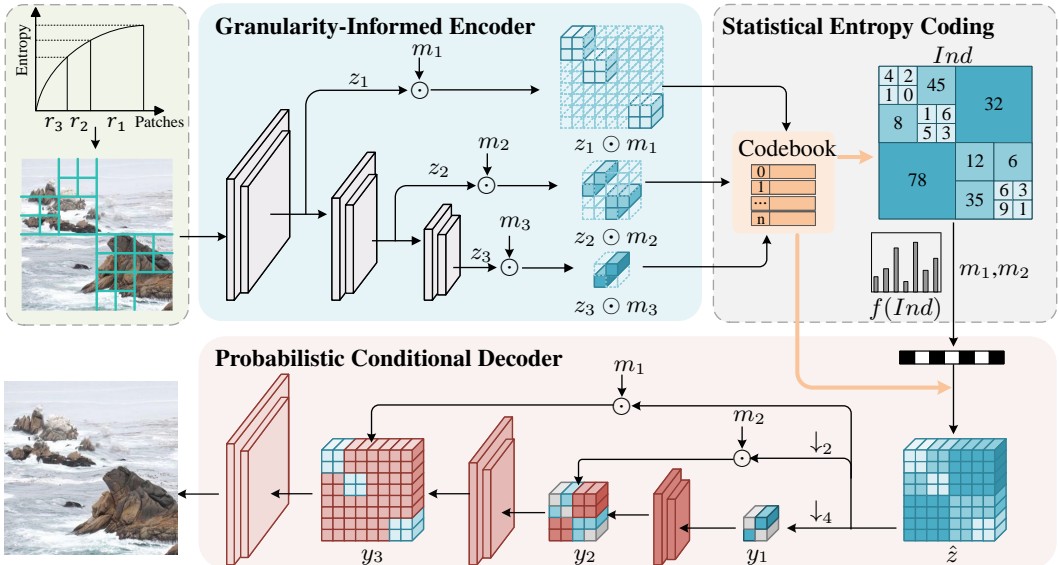

Figure 2: The overall framework of our Control-GIC. In the figure, all components cooperate for efficient compression with end-to-end training, and dashed lines represent the unparameterized entropy coding module. The symbols in the diagram are defined as: $m$: the binary mask; $(\cdot)_\downarrow$: average pooling operation; $\odot$: element-wise multiplication; $f(\cdot)$: the frequency distribution.

context-based trit-plane coding, increasing the R-D performance. In contrast to these methods, this work leverages VQGAN, integrated with classical coding principle (Huffman, 1952), to design a controllable generative compression framework. Our method allows highly flexible and controllable bitrate adaptation while generating plausible results with solely one optimized weight set.

## 3 METHOD

Our goal is to learn a unified generative compression model capable of compressing an image $x$ for flexible and continuous bitrates while ensuring high perceptual fidelity. To this end, we propose Control-GIC, where the overview architecture is illustrated in Figure 2. Control-GIC contains three components: 1) granularity-informed encoder to encode the image into variable-length codes; 2) statistical entropy coding module for bitrate reduction; and 3) probabilistic conditional decoder to reconstruct perceptually plausible results.

### 3.1 GRANULARITY-INFORMED ENCODER

Given an input image $x \in \mathbb{R}^{H \times W \times 3}$, as illustrated in Figure 2, Control-GIC considers the entropy of local patches as the basis of the information density distribution (See Appendix A.2 for proof of correlation) of the image, and divides it into multiple non-overlapped patches sorted by their entropy value from low to high. Then, the granularity-informed encoder distills these patches into hierarchical features of three granularities: fine-grained $z_1 \in \mathbb{R}^{\frac{H}{4} \times \frac{W}{4} \times d}$, medium-grained $z_2 \in \mathbb{R}^{\frac{H}{8} \times \frac{W}{8} \times d}$, and coarse-grained $z_3 \in \mathbb{R}^{\frac{H}{16} \times \frac{W}{16} \times d}$. Supposing there is a target bitrate corresponding to the ratios $(r_1, r_2, r_3) \in [0, 1]$ of three granularities, each ratio specifies the proportion of elements with the lowest entropy to be retained from each $z_i(i = 1, 2, 3)$ and yield binary masks $m_i \in \{0, 1\}$. Here, $m_i$ aligns with the spatial dimensions of $z_i$ to localize the retained elements in $z_i$. This process is executed from coarse to fine, progressively and finely assigning the multi-grained representations.

Subsequently, the determined features in each $\{z_i\}_{i=1}^3$ are matched to the codes in the codebook $C$ and quantized, producing $\hat{z}_i$ and a set of discrete VQ-indices $Ind_i$ that represent the closest matches

in $C$ based on Euclidean distance. This quantization step $\mathbf{q}(\cdot)$ is mathematically formalized as

$$\begin{cases} \hat{z}_i = \mathbf{q}(z_i) = \underset{c_k \in C}{\arg\min} \|z_i - c_k\|, \\ Ind_i = k. \end{cases} \tag{1}$$

where $c_k$ denotes the $k$-th code in the codebook. With three quantized counterparts $\{\hat{z}_i\}_{i=1}^3$, we can construct the hybrid representation $\hat{z}$ to match the spatial scale of the finest granularity as follows:

$$\hat{z} = (\hat{z}_1 \odot m_1) + (\hat{z}_2 \odot m_2)_{\uparrow_2} + (\hat{z}_3 \odot m_3)_{\uparrow_4}, \tag{2}$$

where $(\cdot)\uparrow_4$ and $(\cdot)\uparrow_2$ signify upsampling operations that amplify the spatial dimensions by factors of 4 and 2, respectively. We use nearest neighbor interpolation as it employs replicates values of feature points along both the width and height, ensuring the preservation of the original local structure integrity for each feature point. $\odot$ is element-wise multiplication along the spatial dimension.

## 3.2  PROBABILISTIC CONDITIONAL DECODER

Based on the VQGAN pattern, our decoder receives the indices of $\hat{z}_i$ and correspond masks $m_i$ from the encoder, to reconstruct the features of the encoder by searching for the codebook. However, directly feeding $\hat{z}$ into the decoder layers is sub-optimal as many non-linear transformations in the decoder can cause information loss. While the upsampled components $\hat{z}_2$ and $\hat{z}_3$ maintain their local structure through direct value duplication (Eq. (2)), their global structure is inevitably changed.

To address this problem, we introduce a probabilistic conditional decoder, which formalizes the reconstruction through the conditional probability. Specifically, two downsampling operations $(\cdot)\downarrow_4$ and $(\cdot)\downarrow_2$ are first employed to downscale $\hat{z}$ back to the $\{\hat{z}_i \odot m_i\}_{i=1}^3$ losslessly. We provide $(\hat{z})_{\downarrow_4}$ as the initial decoder input $y_1$ which contains the same $\hat{z}_3 \odot m_3$ as the encoder output to ensure the accuracy of the input. $\hat{z}_1 \odot m_1$ and $\hat{z}_2 \odot m_2$ are provided as conditions to $y_2$ and $y_3$, respectively. These conditions serve as additional guidance for the reconstruction process:

$$\begin{aligned} y_2 &\sim p(y_2 \mid y_1, (\hat{z})_{\downarrow_2} \odot m_2), \\ y_3 &\sim p(y_3 \mid y_2, y_1, \hat{z} \odot m_1). \end{aligned} \tag{3}$$

Consequently, the decoder $D$ begins with the $(\cdot)\downarrow_4$ operation on $\hat{z}$ to produce $y_1$ that is fed into the first decoder layer $D_1$ to generate $y_2$. Then, $y_2 \odot m_2$ are deliberately replaced with the medium-grained representation $(\hat{z})_{\downarrow_2} \odot m_2$ (equal to $\hat{z} \odot m_2$). After that, $D_3$ condition with the $\hat{z} \odot m_1$ and replace the unexact $y_3 \odot (1 - m_1)$, ensuring the precision of features in deep layers:

$$\begin{aligned} y_2 &= D_1(y_1) \odot (1 - m_2) + (\hat{z})_{\downarrow_2} \odot m_2, \\ y_3 &= D_2(y_2) \odot (1 - m_1) + \hat{z} \odot m_1. \end{aligned} \tag{4}$$

This systematic replacement of representations at varying granularities with increasingly precise conditions progressively refines the latent space representation, which facilitates the decoder to diminish information loss and substantially elevates the accuracy of the reconstructed images. It should be noted that, compared to conventional compression methods (Ballé et al., 2017; Mentzer et al., 2020), our method effectively avoids the information loss between the encoder and decoder features except the quantization, thereby achieving nearly minimal-loss reconstruction.

## 3.3  STATISTICAL ENTROPY CODING STRATEGY

As analyzed in Sec. 3.2 and Eq. (4), the decoding process requires bitstreams comprising the features of three granularities and their corresponding masks. Since $\hat{z}_i$ can be retrieved by searching from the codebook based on the indices $Ind_i$, what we need actually are the $\{Ind_i\}_{i=1}^3$ and the masks $\{m_1, m_2\}$. These elements can be encoded via lossless encoding algorithms as they are integers. Specifically, the mask consists of 0 and 1 which can be encoded directly into a binary stream. As for indices, one promising solution is the prefix Huffman coding algorithm (Huffman, 1952), which can generate the shortest average code length for a given symbol set. However, this advantage often comes from the frequency statistics of each element. In this work, directly applying this algorithm is suboptimal as it needs to count the indices frequency used in each independent image and acquire a codebook that maps the indices to the binary codes. This can introduce significant bit overhead

when reconstructing the image based on the Huffman codebook during the decoding. A simplified approach is to treat all indices equally, *i.e.*, assuming the frequency in the codebook is uniform. Nevertheless, the indices, which point to codebook entries, exhibit an uneven frequency distribution, with a minority of codes used for quantization (Zhang et al., 2023).

To address this problem, we introduce a statistical entropy coding strategy that captures the frequency distribution of indices usage across a natural dataset during training. Each index is initialized with a frequency count of 0, and the frequency is updated each time the index is matched during vector quantization. Here, we utilize the frequency statistics at the endpoint of the training process to construct a frequency table tailored for Huffman coding as it is more stable and close to the overall data distribution. We denote the bitrate after coding as $\mathcal{R}(\cdot)$, then the total bitrate of the entire image can be formulated as:

$$\mathcal{R} = \sum_{i=1}^{3} \mathcal{R}(Ind_i) + \mathcal{R}(m_1) + \mathcal{R}(m_2). \tag{5}$$

Note that our model does not optimize network parameters for a specific bitrate. During inference, we control the bitrate using different multi-granularity allocation ratios. For example, we can set a group of hyperparameters $\{r_1, r_2, r_3\}$ to represent the allocation ratios of the masks $\{m_1, m_2, m_3\}$ at three granularities. Since the bitrate depends solely on the granularity representations of local patches, we can flexibly determine the granularities based on the entropy values of local patches, achieving dynamic adaptation in a target of the quality-bitrate adaptive manner in a unified model without any post-training.

### 3.4 LOSS FUNCTION

In our experiments, the overall loss function $\mathcal{L}$ for training Control-GIC contains the loss associated with the VQVAE architecture and GAN component in VQGAN (Esser et al., 2021). The optimization objective of $\mathcal{L}_{\text{VQVAE}}(E, G, C)$ is two-fold: 1) minimizing the distortion between the original inputs $x$ and their reconstructions $\hat{x}$, and 2) constrain the divergence between the continuous representations $z = E(x)$ and their quantized versions $\hat{z}$, as follows:

$$\mathcal{L}_{VQVAE}(E, G, C) = d(x, \hat{x}) + d(z, \hat{z})$$
$$= (d_M + d_P)(x, \hat{x}) + \|sg[z] - \hat{z}\|_2^2 + \|sg[\hat{z}] - z\|_2^2, \tag{6}$$

where we use MSE ($d_M$) and LPIPS ($d_P$) to measure the reconstruction distortion. $sg[\cdot]$ denotes the stop-gradient operation widely utilized in VQ-based models to overcome the non-differentiable nature of quantization. $sg[\cdot]$ enables the quantized representations $\hat{z}$ to propagate gradients directly for optimizing the codebook $C$ and allows the continuous representations $z$ to receive gradients for the refinement of the encoder $E$. Our Control-GIC deviates from the conventional R-D optimization paradigm with no-parametric indices compression. This enables the model to adapt flexibly to various data types and quality levels, transcending the fixed R-D trade-off in most generative methods.

For GAN loss, we use the patch-based discriminator in (Isola et al., 2017) to differentiate between original and compressed images:

$$\mathcal{L}_{GAN}(\{E, G, C\}, D) = \mathbb{E}_{x \sim p(x)}[\log D(x) + \log(1 - D(G(\hat{z})))]. \tag{7}$$

Therefore, the total loss $\mathcal{L}$ is formulated by

$$\mathcal{L} = \mathcal{L}_{VQVAE} + \lambda \mathcal{L}_{GAN}. \tag{8}$$

We use the hyperparameter $\lambda$ to control the trade-off between VQVAE and GAN losses.

## 4 EXPERIMENT

### 4.1 EXPERIMENTAL SETUP

Our method is based on MoVQ (Zheng et al., 2022) which improves the VQGAN model by adding spatial variants to representations within the decoder, avoiding the repeat artifacts in neighboring patches. We leverage the pre-trained codebook in MoVQ and carefully redesign the architecture.

**Training & Inference**. We randomly select 300K images from the OpenImages (Krasin et al., 2017) dataset as our training set, where the images are randomly cropped to a uniform $256 \times 256$ resolution.

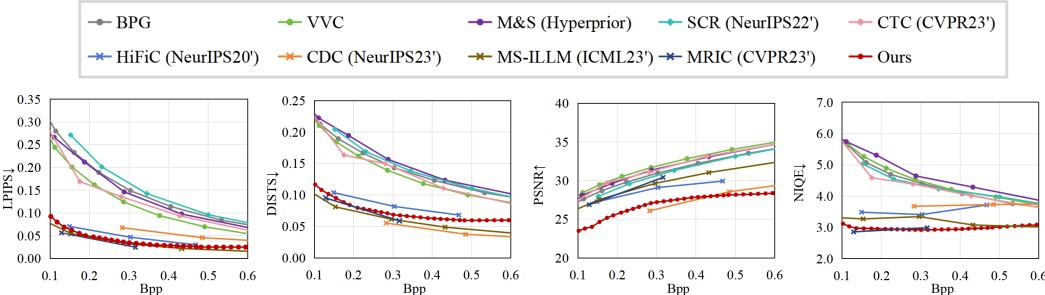

Figure 3: Comparisons with existing methods on the Kodak. The lines with forks represent generative compression methods and the lines with rhombus represent variable-rate and progressive methods.

Within our model, we take three representation granularities: $4 \times 4$, $8 \times 8$, and $16 \times 16$. The codebook $C \in \mathbb{R}^{k \times d}$ comprises $k = 1024$ code vectors, each with a dimension of $d = 4$. We train the model for 0.6M iterations with the learning rate of $5 \times 10^{-5}$ on NVIDIA RTX 3090 GPUs. Throughout the training, we maintain the ratio setting of $(50\%, 40\%, 10\%)$ for the fine, medium, and coarse granularity, respectively. During inference, our Control-GIC can process images of any resolution and allow fine bitrate adjustment using a unified model.

**Evaluation**. We evaluate our method on the Kodak (Kodak, 1993), DIV2K (Agustsson & Timofte, 2017), and CLIC2020 (Toderici et al., 2020) datasets. Kodak contains 24 high-quality images at $768 \times 512$ resolution. DIV2K and CLIC2020 contain 100 and 428 high-resolution images, respectively, with resolutions extending up to 2K. We carry out multi-dimensional evaluation and utilize a comprehensive set of evaluation metrics including perceptual metrics: **LPIPS** (Zhang et al., 2018), **DISTS** (Ding et al., 2020), distortion metric: **PSNR**, generative metrics: **FID** (Heusel et al., 2017), **KID** (Bińkowski et al., 2018), as well as the no-reference measurement: **NIQE** (Mittal et al., 2012) to thoroughly assess the performance of our method. More details are in the Appendix A.1.

## 4.2 PERFORMANCE COMPARISON

We compare the proposed Control-GIC with four types of state-of-the-art (SOTA) image compression methods: **(1) Generative compression methods.** HiFiC (Mentzer et al., 2020) and MRIC (Agustsson et al., 2023) leverages conditional GAN to purse the rate-distortion-perception trade-off. MS-ILLM (Muckley et al., 2023) improves statistical fidelity using local adversarial discriminators. CDC (Yang & Mandt, 2024) is a representative diffusion-based lossy image compression approach; **(2) Variable-rate and progressive compression methods**. SCR (Lee et al., 2022b) proposes a 3D important map adjusted by quality level to select the representation elements for variable rates. CTC (Jeon et al., 2023a) progressively decodes the bit stream truncated at any point to regulate the bitrate; **(3) Classical NIC method**. M&S (Hyperprior) (Ballé et al., 2018) trains separate models for different bitrates; **(4) Traditional codecs** BPG (Bellard) and VVC (VTM10.0) (Bross et al., 2021).

**R-D Performance**. In Figure 3, we first provide the R-D curves for all methods, evaluated using four metrics: LPIPS, DISTS, PSNR, and NIQE, on the Kodak dataset. Our Control-GIC surpasses most methods across 4 distinct metrics and achieves comparable performance with the most state-of-the-art methods MS-ILLM and MRIC in LPIPS and NIQE, even though they are trained separately for specific R-D points. Compared to SCR and CTC, Control-GIC achieves finer granularity flexibility for bitrate control while preserving obvious preferable perceptual quality. Besides, since conventional CNN-based NIC methods, *e.g.* M&S (Hyperprior) and SCR, optimize for the R-D trade-off using pixel-wise MSE loss, it can be observed that they produce relatively higher PSNR than generative methods.

Then, we conduct comparisons on the DIV2K (Agustsson & Timofte, 2017) dataset. As illustrated in Figure 4, in addition to the four metrics in Figure 3, we include FID and KID to provide a more comprehensive evaluation. The results demonstrate that our Control-GIC maintains promising competitiveness against existing methods in almost all the metrics across a wide spectrum of bitrates. We also investigate the effectiveness of our method on the CLIC2020 dataset, where the results are provided in Figure 10 (see Appendix A.5).

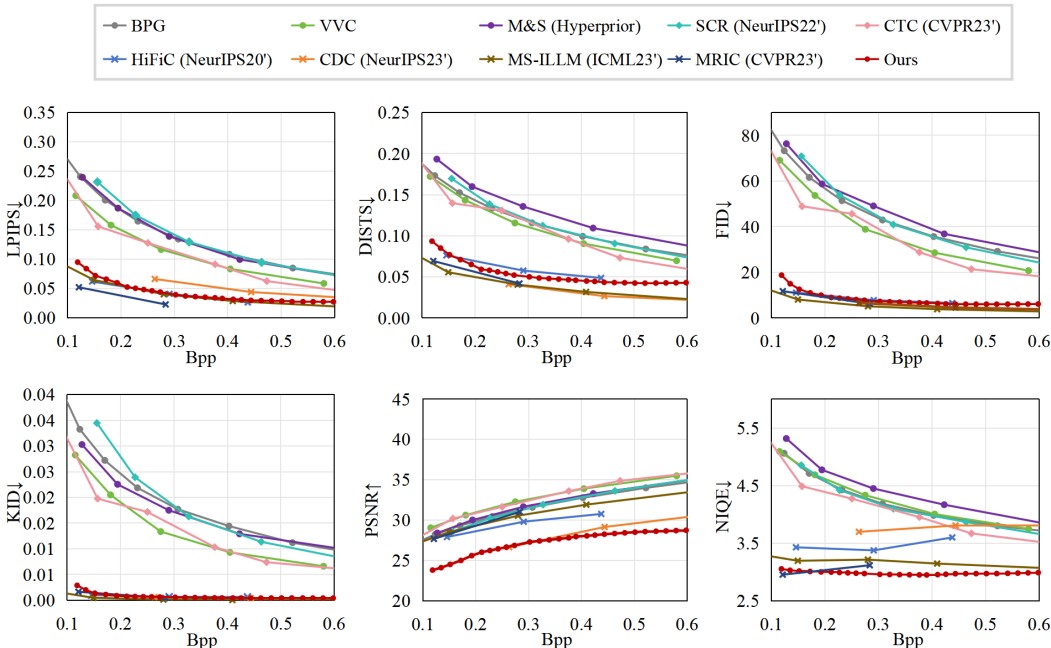

Figure 4: Compression performance on the DIV2K with compared methods. The lines with forks represent GIC methods, and the lines with rhombus represent variable-rate and progressive methods.

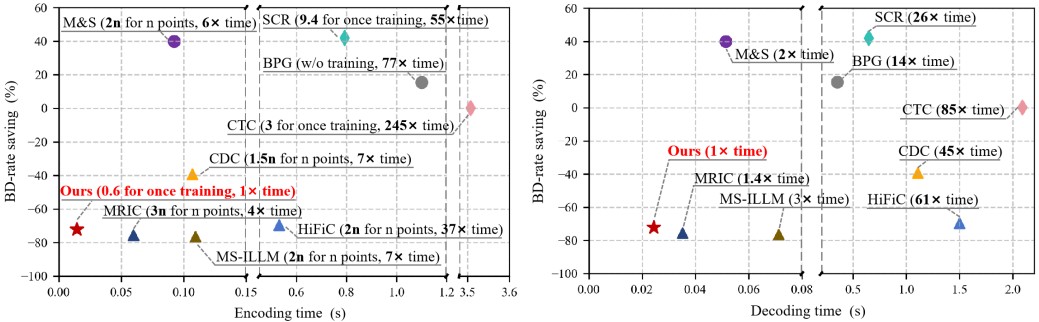

Figure 5: Comparison of model efficiency for all methods based on four terms: encoding time (s), decoding time (s), BD-rate saving (%), and training steps (M) on the Kodak dataset. The diamond icons represent variable-rate and progressive methods, and the triangles represent GIC methods, with the number of iterations required for model training and the encoding/decoding time multiplier compared to our method labeled in parentheses after each method name.

**Model Efficiency**. To demonstrate the efficiency of the proposed method, in Figure 5, we analyze existing state-of-the-art methods and our Control-GIC on four terms: encoding time (sec.), decoding time (sec.), BD-rate saving (%) and training steps (M). For fair comparisons, all the methods are evaluated using their original public code and pre-trained models on the same NVIDIA 3090 GPU. We calculate the average encoding and decoding time on the Kodak dataset. The BD-rate saving is evaluated by quantifying the Bpp-LPIPS results, using VVC as the anchor. As shown in Figure 5, MRIC, MS-ILLM, and Control-GIC obtain very close BD-rate saving and are superior to others. For the inference speed, M&S, CTC, and HiFiC suffer from critical time costs in both encoding and decoding. CDC applies a lightweight diffusion variational autoencoder, which benefits the encoding process but struggles with more decoding time due to its iterative reverse process. Our method achieves the fastest encoding/decoding time, which is $7\times$ faster than MS-ILLM and $4\times$ faster than MRIC in encoding, and $3\times$ faster than MS-ILLM and $1.5\times$ faster than MRIC in decoding. Moreover, the single-point training methods (*e.g.*, M&S, HiFiC, CDC, MRIC, MS-ILLM) require independent

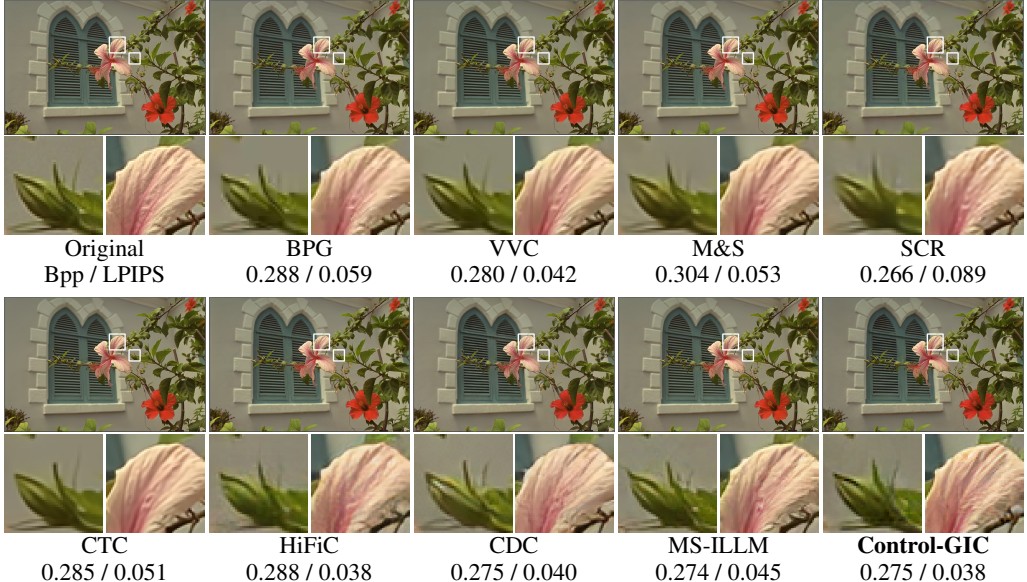

| Original | BPG | VVC | M&S | SCR |
| Bpp / LPIPS | 0.288 / 0.059 | 0.280 / 0.042 | 0.304 / 0.053 | 0.266 / 0.089 |
| CTC | HiFiC | CDC | MS-ILLM | **Control-GIC** |
| 0.285 / 0.051 | 0.288 / 0.038 | 0.275 / 0.040 | 0.274 / 0.045 | 0.275 / 0.038 |

Figure 6: Reconstructed Kodak images by existing state-of-the-art methods and our Control-GIC.

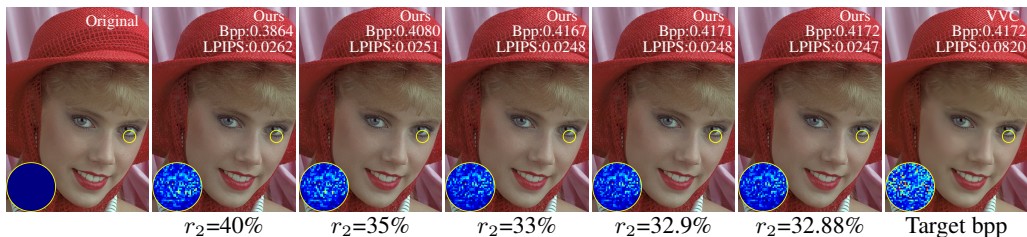

| $r_2$=40% | $r_2$=35% | $r_2$=33% | $r_2$=32.9% | $r_2$=32.88% | Target bpp |

Figure 7: The fine-grained control over the bitrate by Control-GIC. As $r_2$ diminishes, the proportion of fine-grained features correspondingly increases, leading to a higher total count of codes, and a consequent increase in the bitrate (measured in bpp) and reconstruction quality (the lower left corner visualizes the difference maps between the original image and reconstructed ones).

training of $n$ models for $n$ R-D points. The proposed model requires only a single training session that enables compression across various bitrates, with the total training steps being substantially reduced to 0.6 million steps. Although SCR and CTC support multiple compression rates in a single model, they still involve many more training steps, especially SCR which is more than $15\times$ that of ours. By comparison, our Control-GIC can achieve a promising balance among training costs, inference speed, and BD-rate saving.

**Qualitative Comparison**. In Figure 6, we visualize the reconstructed images by all compared methods. As we can see, VVC, M&S, SCR, and CTC produce the results with noticeable blurs and artifacts. While HiFiC, CDC, and MS-ILLM can yield clearer details, their images contain some misleading textures and artifacts not present in the original ones. By comparison, our method excels at preserving texture integrity and image sharpness. More visual results are in Appendix A.7.

## 4.3 ABLATION STUDY

**Fine-grained Control of Bitrate**. As analyzed in Sec. 3.1, the proposed **Control-GIC** can flexibly control the bitrates through the granularity ratios $(r_1, r_2, r_3)$. To validate the effects, in Figure 7, we visualize the reconstructed samples of our method, where VVC is adopted as a reference. We fix the ratio of the coarse-grained features as $r_3 = 50\%$ and adjust the ratio of medium-grained features $r_2$, thus one can directly obtain the ratio $r_1 = 1 - r_2 - r_3$ for the fine-grained features. It can be observed that, as $r_2$ diminishes, $r_1$ increases, resulting in a higher total count of codes and, consequently, a

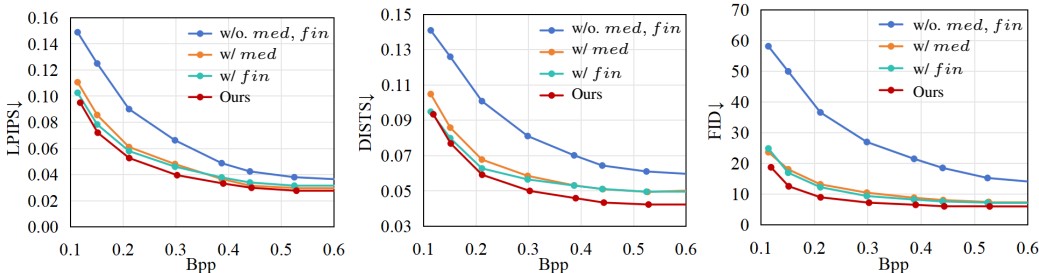

Figure 8: The ablation experiments on multi-grained conditions for the probabilistic conditional decoder. All experiments are performed on the DIV2K dataset.

Table 1: Bit cost comparison between our statistical entropy coding and Huffman coding with uniform frequency for each index on the Kodak dataset.

| Granularity Ratio | Huffman $w$. Uniform Frequency | Statistical Entropy Coding (Ours) | Bit Saving (%) |
|---|---|---|---|
| 100%, 0, 0 | 0.625 | 0.594 | 5.0% |
| 60%, 30%, 10% | 0.445 | 0.432 | 2.9% |
| 20%, 50%, 30% | 0.235 | 0.234 | 0.4% |

higher bitrate (measured in bpp), where the perceptual quality gradually improves. Furthermore, the continuous controllable compression in Figure 7 indicates that our method enables precise regulation of the bpp, allowing for minute adjustments within a range as narrow as 0.001 (as exemplified by the change from 0.4171 in the third-to-last column to 0.4172 in the second-to-last column).

**Impacts of Probabilistic Conditional Decoder.** In this work, we propose the probabilistic conditional decoder which formalizes the reconstruction through the conditional probability. Here, we investigate the contributions of the conditions: the medium-grained $(\hat{z})_{\downarrow_2} \odot m_2$ (denoted as $med$) and fine-grained $\hat{z} \odot m_1$ (denoted as $fin$) to the decoder in Eq. (4) to reveal the impacts of the proposed probabilistic conditional decoder, where their R-D performance is illustrated in Figure 8. We can observe that the model without the $med$ and $fin$ produces the worst results in three metrics. Besides, $med$ and $fin$ present a significant improvement in model performance, and conditioning upon both presents the best results, especially on DISTS. Moreover, the results also validate that adding the fine-grained $fin$ to the model brings more benefits than adding $med$. This is because the fine-grained $fin$ can correct the features in deeper layers of the decoder, thereby improving the accuracy after multiple non-linear transformations within the decoder.

**Efficiency of Statistical Entropy Coding.** In Table 1, we compare the proposed statistical entropy coding strategy to Huffman coding with uniform frequency for each index. The superiority of our statistical entropy coding strategy becomes increasingly evident as more codes are employed (*i.e.*, higher bpp) in the coding process. Notably, at the highest bpp, our approach achieves nearly 5.0% bit saving than Huffman coding with uniform frequency distribution thereby verifying its efficiency.

## 5 CONCLUSION

In this work, we propose Control-GIC, an innovative controllable generative image compression framework that addresses the critical challenge of flexible rate adaptation By leveraging a VQGAN foundation and correlating local image information density with granular representations, Control-GIC achieves fine-grained bitrate control across a wide range while maintaining high-fidelity compression performance. We propose a granularity-informed encoder that represents the image patches of sequential spatially variant VQ-indices to support precise variable rate control and adaptation. Then, a non-parametric statistical entropy coding is devised to encode the VQ-indices losslessly. In addition, we develop a probabilistic conditional decoder, which aggregates historic encoded multi-granularity representations to reconstruct hierarchical granular features in a conditional probability manner, achieving realism improvements. Experiments validate the superior effectiveness, compression efficiency, and flexibility of our method.

ACKNOWLEDGMENTS

This work was supported in part by National Natural Science Foundation of China (No. 62331003, 62120106009, 62302141, 62471278), Beijing Natural Science Foundation (L223022), the Fundamental Research Funds for the Central Universities (JZ2024HGTB0255), and the Taishan Scholar Project of Shandong Province under Grant tsqn202306079.

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

## A APPENDIX

### A.1 EVALUATION METRICS

We adopt a comprehensive set of evaluation metrics to thoroughly assess the performance of our image compression and reconstruction techniques. Our selection encompasses perceptual metrics, distortion metrics, generative metrics, and a no-reference metric, ensuring a multifaceted evaluation. The perceptual metrics include LPIPS (Learned Perceptual Image Patch Similarity) (Zhang et al., 2018), which measures the perceptual difference between images, and DISTS (Deep Image Structure and Texture Similarity) (Ding et al., 2020), which evaluates the structural dissimilarity. These metrics are crucial for understanding how closely compressed and reconstructed images resemble their original counterparts in terms of human visual perception. We also include the widely recognized distortion metric PSNR (Peak Signal-to-Noise Ratio), which quantifies the pixel-level differences between the original and reconstructed images. PSNR is a standard in the field, providing a straightforward measure of image fidelity. For generative metrics, we employ FID (Fréchet Inception Distance) (Heusel et al., 2017) and KID (Kernel Inception Distance) (Bińkowski et al., 2018) to offer statistical assessments of the similarity between the distributions of original images and those of reconstructed images, which is particularly valuable in the context of generative models. NIQE (Natural Image Quality Evaluator) (Mittal et al., 2012) stands out as a no-reference metric, capable of evaluating image quality without requiring an original reference image. This feature renders NIQE exceptionally beneficial in applications such as super-resolution where an original high-resolution image may not be available. For comparison with other methods on FID and KID, we divide the DIV2K dataset into 6,573 patches, and the CLIC2020 dataset into 28,650 patches, each of size 256.

### A.2 CORRELATION BETWEEN ENTROPY AND INFORMATION DENSITY

Inspired by Celik (Celik, 2014), we measure the information density of image regions based on a non-parametric spatial entropy algorithm. Unlike feature-level entropy models based on neural networks (Ballé et al., 2017; 2018), our Control-GIC does not rely on a neural entropy model and is not optimized for entropy during training. Instead, we adopt a non-parametric algorithm to reduce computational overhead while maintaining robust performance in granularity selection. The mathematical formulation of this process is described below.

Consider a pixel $x \in \Omega$ with a value $p_x$ within the interval [-1,1], where $\Omega$ denotes a patch of the image. We define the bin of each pixel value as $i = -1 + \frac{2k}{n-1}$, where $k = 0, 1, 2, ..., n - 1$, and the number of bins $n$ is set to 32. The Gaussian distance $f_{x,i}$ between the pixel $x$ and each bin $i$ is computed as:

$$f_{x,i} = \exp\left(-\frac{(p_x - i)^2}{2\sigma^2}\right), \tag{9}$$

where $\sigma$ is the standard deviation. Thus, $f_{x,i}$ exhibits an unnormalized, truncated discrete Gaussian distribution over the bins $i$. This implies that $f_{x,i}$ models the probability of $p_x$ being associated with bin $i$, reflecting the likelihood of pixel value distribution across bins.

Next, we compute the average of $f_{x,i}$ across all pixels within the patch $\Omega$ to obtain the probability distribution $f_{\Omega,i}$ for this region. This is normalized to yield $\overline{f_{\Omega,i}}$. The average operation is denoted as $\underset{x \in \Omega}{mean}$, and the mathematical expression is as follows:

$$\overline{f_{\Omega,i}} = \frac{f_{\Omega,i}}{\sum_j f_{\Omega,j}}, \quad \text{where } f_{\Omega,i} = \underset{x \in \Omega}{mean} f_{x,i}. \tag{10}$$

Finally, the spatial entropy $H(\Omega)$ of the patch $\Omega$ is computed as:

$$H(\Omega) = -\sum_i \overline{f_{\Omega,i}} \log \overline{f_{\Omega,i}}. \tag{11}$$

Control-GIC considers the spatial entropy of local patches to model the information density distribution of the image. The image is divided into multiple non-overlapping patches, which are sorted by their entropy values from low to high, allowing the assignment of multi-grained features.

### A.3 DETAILS FOR GRANULARITY DIVISION

As illustrated in Figure 2, given an input image $x$ partitioned into a series of patches, the encoder $E$ first computes the entropy values for all patches and sorts them in ascending order. We categorize these patches into three distinct granularity levels: fine, medium, and coarse. Upon completion of training, an index frequency table is generated, which facilitates the assignment of codes to each index during the entropy encoding process. Through empirical analysis, for a codebook comprising 1024 codes, the average code length of all indices is determined to be $L = 10.3875$. For an input image of size $H \times W$, the range of assigned indices spans from $\frac{H}{16} \times \frac{W}{16}$ to $\frac{H}{4} \times \frac{W}{4}$, corresponding to fully coarse-grained and fine-grained patch divisions, respectively. For any given combination of granularity ratios $(r_1, r_2, r_3)$, the theoretical bit-per-pixel (bpp) values for indices and mask are derived as follows:

$$Bpp_{Indices} = \frac{L}{256} \left( 16r_1 + 4r_2 + r_3 \right), \tag{12}$$

$$Bpp_{Mask} = \frac{1}{256} \left( 4r_1 + r_2 \right), \tag{13}$$

where $r_1$, $r_2$, and $r_3$ represent the ratios of fine, medium, and coarse patches, respectively.

Based on these equations, a query table mapping granularity ratios to theoretical bpp values is constructed. When a user-specified bpp is provided, the model automatically searches for the closest theoretical bpp in the query table and assigns the corresponding granularity ratios. In our experiments, the discrepancy between the theoretical and actual bpp values is consistently less than 0.05. Specifically, at high bit rates, the actual bpp tends to be slightly lower than the theoretical value, whereas at low bit rates, the actual bpp is marginally higher. To address this, our model allows users to adjust the granularity ratios to minimize the error and achieve optimal results. Table 2 provides a simplified query table illustrating the relationship between granularity ratios and theoretical bpp values. This approach enables precise control over bitrates, ensuring high flexibility and accuracy in compression.

Table 2: Partial query table of granularity ratios and bpp values.

| Granularity Ratio | | | Bpp |
| --- | --- | --- | --- |
| $r_1$ | $r_2$ | $r_3$ | |
| 0 | 23% | 77% | 0.070 |
| 10% | 67% | 23% | 0.187 |
| 37% | 46% | 17% | 0.330 |
| 61% | 30% | 9% | 0.460 |
| 90% | 10% | 0 | 0.616 |

### A.4 VISUALIZATION OF DIFFERENT GRANULARITY RATIOS

In our method, the compression performance is highly dependent on the combination of granularity ratios $r_1$, $r_2$, and $r_3$, which correspond to fine, medium, and coarse patches, respectively. To investigate their impact, we compress images using different combinations of these ratios and visualize the qualitative results on the Kodak dataset in Figure 9 (a). Generally, the visual quality improves as $r_1$ and $r_2$ increase. This is because a higher proportion of fine- and medium-granularity patches provides more local texture cues, which are crucial for detail recovery. However, increasing $r_2$ and $r_3$ also leads to higher bitrate costs.

To further analyze the effect of granularity ratios, we conduct experiments on images containing small faces, comparing our results with those of CDC (Yang & Mandt, 2024) and MS-ILLM (Muckley et al., 2023). As shown in Figure 9 (b), small faces pose a significant challenge for image compression. Both CDC and MS-ILLM struggle to recover fine details in these regions. When coarse- and medium-granularity ratios are excessively high in the face area (i.e., large patch sizes), our model also fails to reconstruct facial details effectively. However, by assigning fine-granularity patches to the face region and reducing their patch size, artifacts are significantly alleviated, resulting in clearer details. For

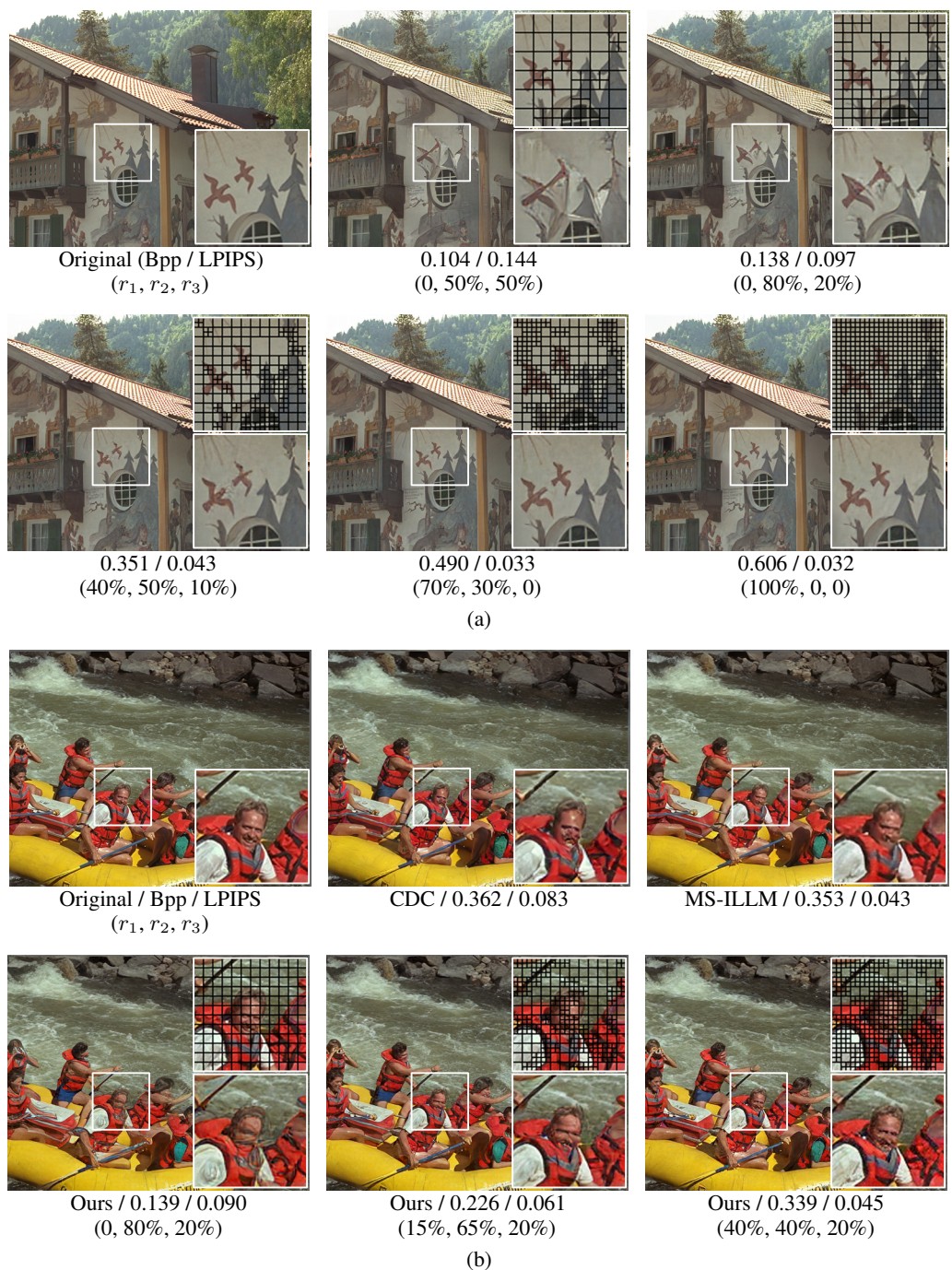

Figure 9: Compression results with different granularity ratios. (a) Building images; (b) Small face images. Here, $r_1$, $r_2$, and $r_3$ denote fine, medium, and coarse granularity ratios, respectively.

instance, our model with granularity ratios (40%, 40%, 20%) achieves better reconstruction quality at a lower bitrate. Despite these improvements, the complexity of small faces highlights the need for more targeted design in future work.

## A.5 EFFECTIVENESS ON THE CLIC2020 DATASET

Here, we evaluate the proposed method on the CLIC2020 dataset (Toderici et al., 2020) which contains 428 images. In Figure 10, we provide the R-D curves for our Control-GIC and existing state-

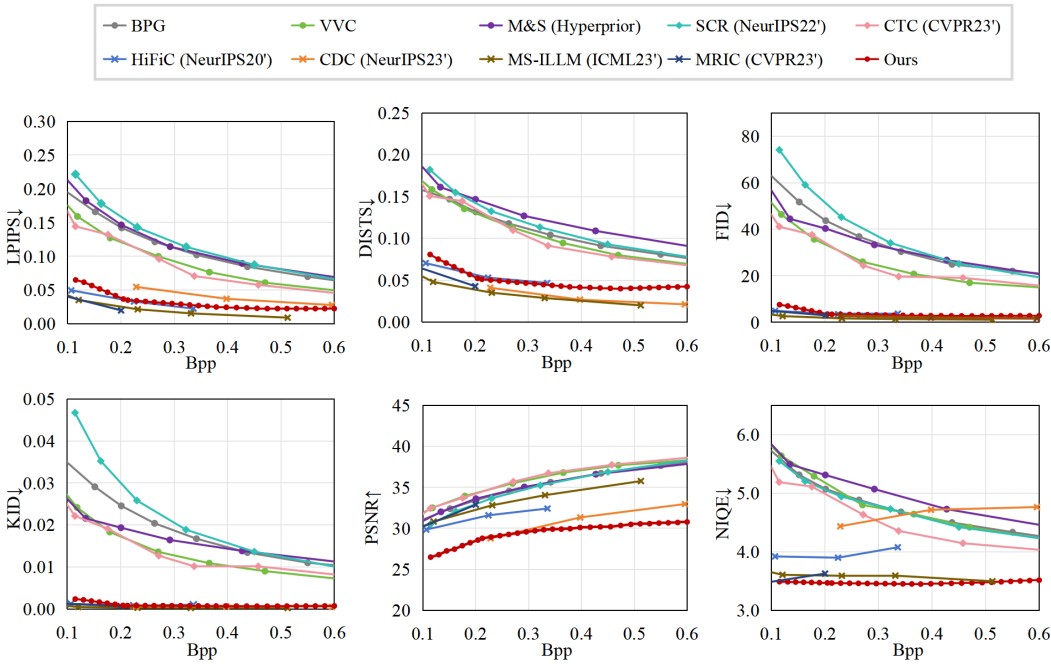

Figure 10: Compression performance on the CLIC2020 dataset with compared methods. The lines with forks represent GIC methods, and the lines with rhombus represent variable-rate and progressive methods.

of-the-art methods, which are measured in six metrics: LPIPS, DISTS, FID, KID, PSNR, and NIQE. It can be seen that Control-GIC achieves superior performance in most metrics over conventional codecs BPG, VVC, and variable-rate and progressive methods SCR, CTC. Compared to generative compression methods which are trained separately for multiple R-D points, our Control-GIC still maintains competitive performance, which validates that our method can achieve optimal trade-off between flexibility and effectiveness.

## A.6 EXTENSION TO EXTREMELY LOW BITRATE COMPRESSION

As described in Section 4, our method take three representation granularities: $4 \times 4$, $8 \times 8$, and $16 \times 16$. The codebook $C \in \mathbb{R}^{k \times d}$ consists of $k = 1024$ codes, each with a dimension of $d = 4$. The lowest bitrate of our method corresponds to a fully coarse-grained partition, *i.e.* $(r_1, r_2, r_3) = (0, 0, 100\%)$. In this section, we investigate the performance of our method for extremely low bitrate compression ($< 0.05$ bpp), comparing it with Mao *et al.* (Mao et al., 2023), a VQGAN-based method designed for very low bitrate compression. As shown in Table 3, our method achieves the best LPIPS on both the Kodak and CLIC2020 datasets while maintaining lower bpp, validating its superiority. Furthermore, Figure 11 provides visual comparisons of compressed images, illustrating that our method generalizes well to extremely low bitrates and produces vivid reconstructions.

Table 3: Quantitative performance at the extremely low bitrate on the Kodak and CLIC2020 datasets.

| Dataset | Kodak | | CLIC2020 | |
|---|---|---|---|---|
| Method | Ours | Mao *et al.* | Ours | Mao *et al.* |
| Bpp↓ | 0.0381 | 0.0391 | 0.0372 | 0.0389 |
| LPIPS↓ | 0.115 | 0.136 | 0.086 | 0.112 |

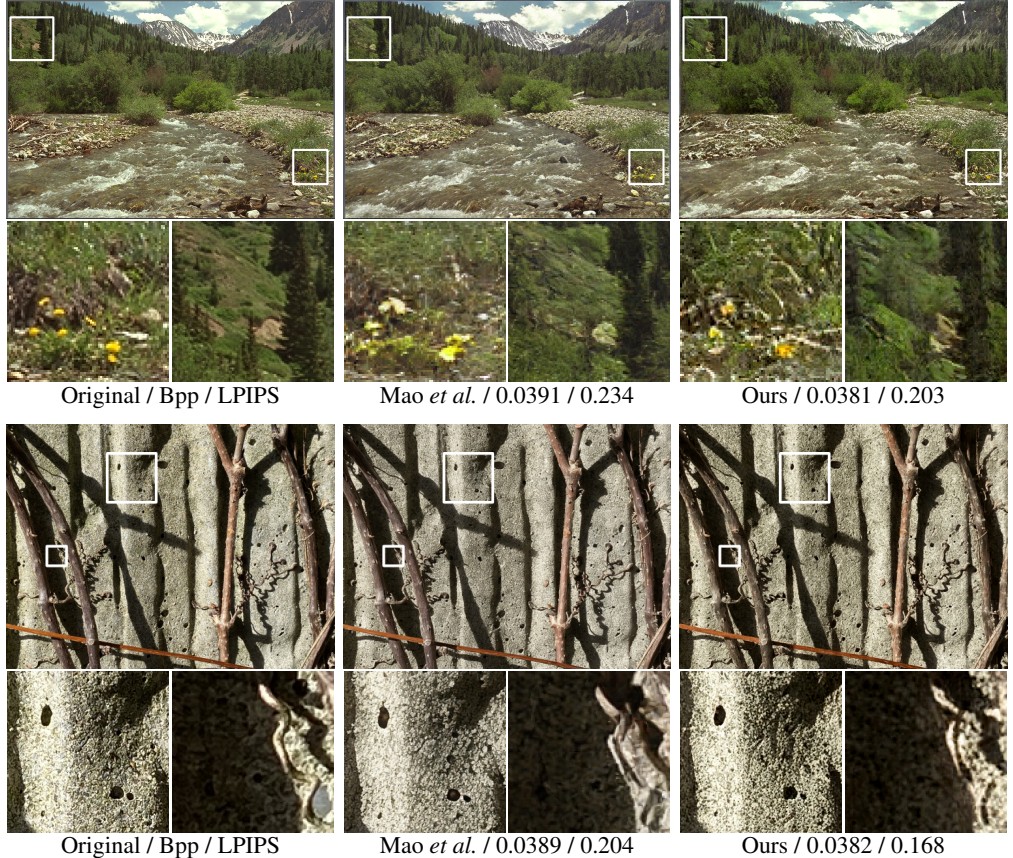

| Original / Bpp / LPIPS | Mao *et al.* / 0.0391 / 0.234 | Ours / 0.0381 / 0.203 |

| Original / Bpp / LPIPS | Mao *et al.* / 0.0389 / 0.204 | Ours / 0.0382 / 0.168 |

Figure 11: The visual results compressed by Mao *et al.* (Mao et al., 2023) and our Control-GIC at the extremely low bitrate on the Kodak (top) and CLIC2020 (bottom) datasets.

## A.7 Additional visualization

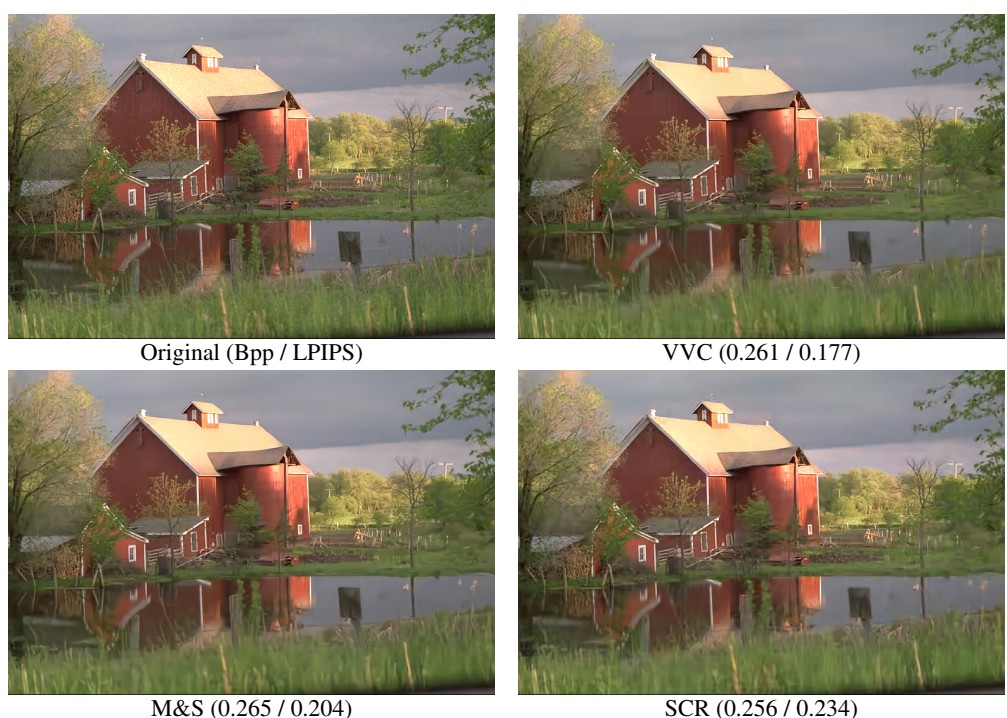

| Original (Bpp / LPIPS) | VVC (0.261 / 0.177) |

| M&S (0.265 / 0.204) | SCR (0.256 / 0.234) |

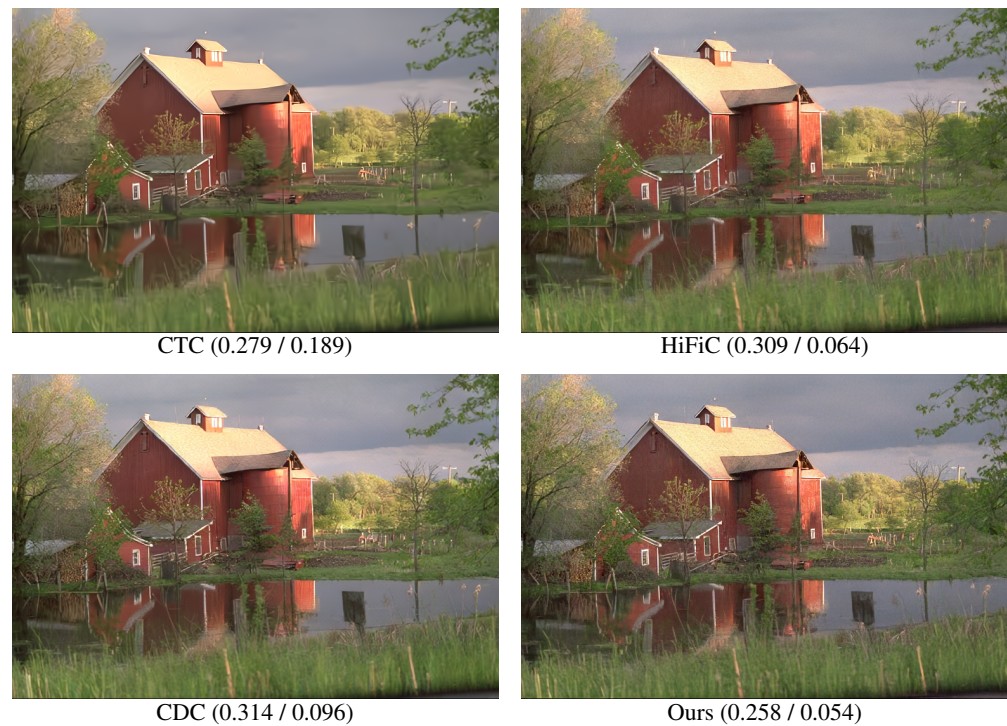

CTC (0.279 / 0.189)          HiFiC (0.309 / 0.064)

CDC (0.314 / 0.096)          Ours (0.258 / 0.054)

Figure 12: Rconstructed images of Kodim22. Bitrate (bpp) and LPIPS are below each image.

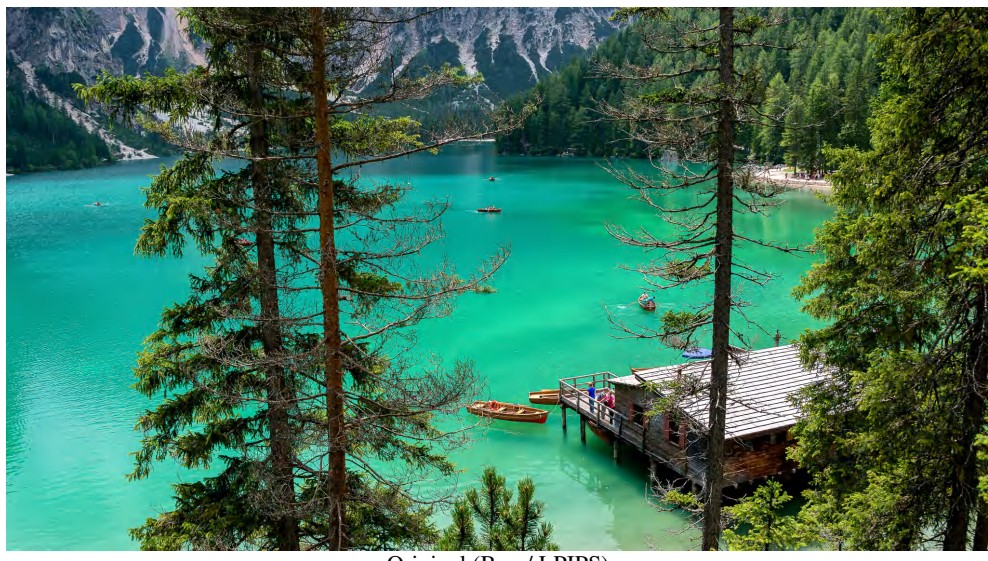

Original (Bpp / LPIPS)

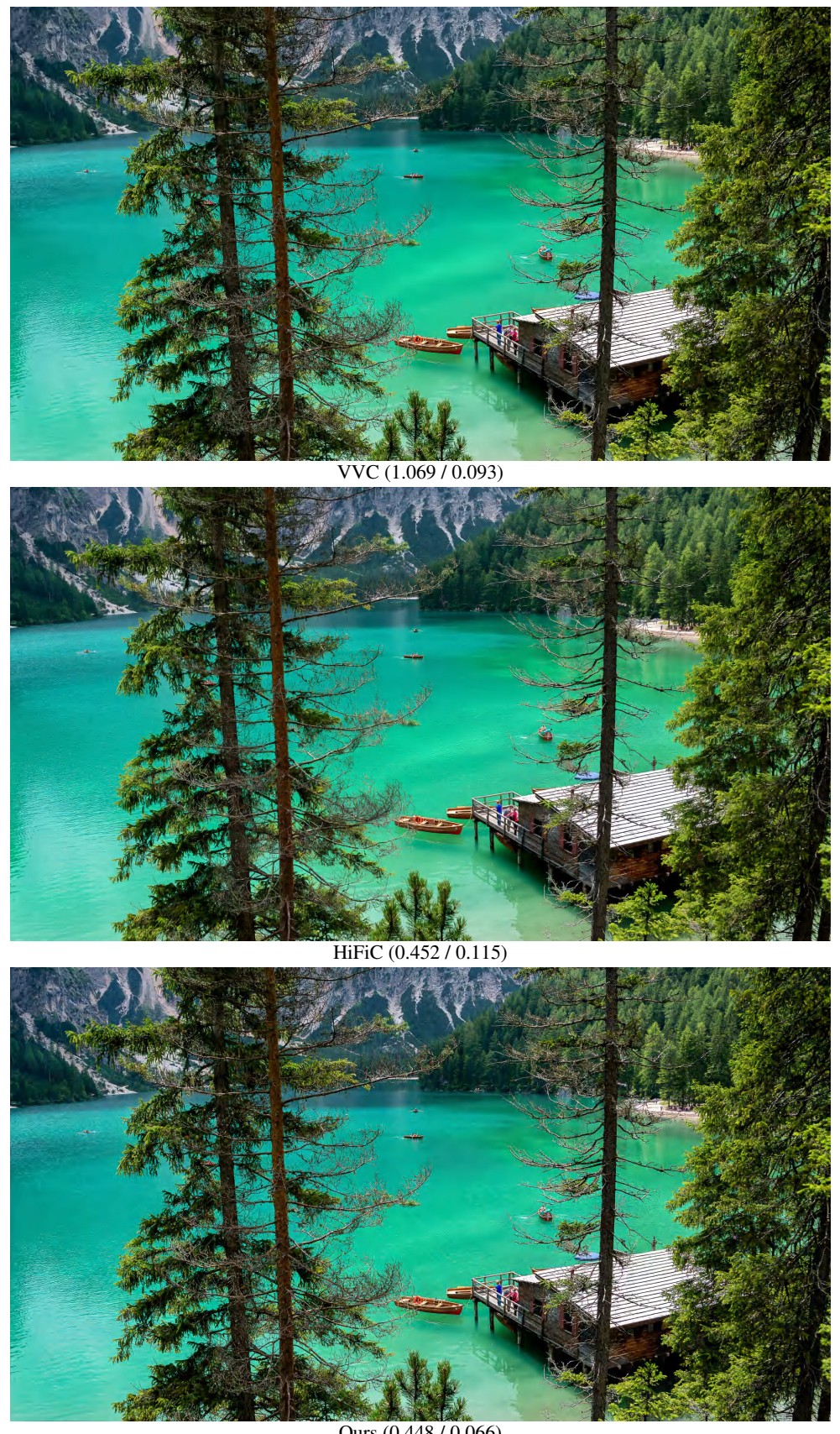

VVC (1.069 / 0.093)

HiFiC (0.452 / 0.115)

Ours (0.448 / 0.066)

Figure 13: Rconstructed images of DIV2K0807. Bitrate (bpp) and LPIPS are below each image.

