# OpenReview forum: "Once-for-All: Controllable Generative Image Compression with Dynamic Granularity Adaptation"
_ICLR.cc/2025/Conference — ICLR 2025 Poster_

### Official Review · Reviewer_bPMe · 2024-10-21

**Soundness:** 3
**Presentation:** 3
**Contribution:** 3
**Rating:** 6
**Confidence:** 5

**Summary:**

This paper presents a novel framework for controllable generative image compression (Control-GIC). The proposed Control-GIC integrates a granularity-aware encoder to enable precise variable rate control and adaptation, a non-parametric statistical entropy coding method for lossless encoding of VQ-indices, and a probabilistic conditional decoder to reconstruct hierarchical granular features. Experimental results on the Kodak and DIV2K datasets demonstrate that Control-GIC not only delivers strong perceptual compression performance but also provides highly flexible and controllable bitrate adaptation.

**Strengths:**

1. A comprehensive explanation on adapting VQ-GAN into a rate-adaptive perceptual codec.

2. Highly flexible and controllable bitrate adaptation.

3. Well-conducted ablation study.

4. Open-sourced code! This is fantastic, and I want to thank the authors for their commitment to sharing this.

**Weaknesses:**

1. Lack of comparison with the latest state-of-the-art (SOTA) perceptual codec, MS-ILLM [1]. It appears that MS-ILLM outperforms both HiFiC and Control-GIC in terms of compression performance.

2. Control-GIC utilizes a mask for multi-scale encoding, but the method for generating the masks $m_1, m_2$ and $m_3$ during training and inference is not clearly explained. Please provide a more detailed description of this process.

3. Misuse of capitalization: 'The Encoding and decoding' on line 432 should be corrected to 'The encoding and decoding.'"

[1] MJ Muckley, A El-Nouby, K Ullrich, H Jegou, J Verbeek. Improving Statistical Fidelity for Neural Image Compression with Implicit Local Likelihood Models. In ICML, 2023.

**Questions:**

Please see the above weakness.

**Details Of Ethics Concerns:**

There are no ethics concerns.

---

> ### Author Response · Authors · 2024-11-21
> **Response to Reviewer bPMe**
>
> **Weakness 1**: Comparison with MS-ILLM
>
> **Response 1**: We apologize that the comparison with MS-ILLM is missing. In the revised manuscript, we have compared with MS-ILLM [ref1] on both Kodak and DIV2K datasets, where the results are in **Figure 3 and Figure 4 (revised manuscript, $\textcolor[RGB]{65, 105, 225}{highlighted\ in\ blue}$)**, respectively. One can see that, MS-ILLM is trained for specific R-D points, which achieve relatively higher quantitative performance in LPIPS and DISTS. However, even though, our method exhibits comparable performance in FID and KID, and better NIQE than MS-ILLM. Here, for simple clarity, we provide training steps (M), BD-rate (BD-LPIPS) results on the DIV2K dataset, and encoding/decoding time (s) on the Kodak dataset for quick comparison using VVC as an anchor, as follows:
> | Method | Training Steps (M) | BD-rate (BD-LPIPS)↓ | Encoding time (s) | Decoding time (s) |
> |:-------:|:-------:|:-------:|:-------:|:-------:|
> | BPG | - | 34.5704 | 1.1001 | 0.3514 |
> | M&S | 2n | 33.4252 | 0.0923 | 0.0514 |
> | SCR | 9.4 | 38.9885 | 0.7924 | 0.6477 |
> | CTC | 3 | -7.8118 | 3.5067 | 2.0880 |
> | HiFiC | 2n | -72.6776 | 0.5306 | 1.4993 |
> | CDC | 1.5n |-48.2543 | 0.1069 | 1.1054 |
> | MS-ILLM | 2n |-74.0735 | 0.1094 | 0.0713 |
> | Ours | **0.6** | -67.5585 | **0.0143** | **0.0244** |
>
> Our method shows the fastest encoding/decoding time, which is $7\times$ faster than MS-ILLM in encoding, and $3\times$ faster than MS-ILLM in decoding. The comparisons demonstrate that our method can achieve a better trade-off between efficiency and effectiveness while preserving high flexibility to support variable bitrates in one model.
>
>
> **Weakness 2**: Explanation on the obtaining for masks $m_1$, $m_2$, $m_3$
>
> **Response 2**: In this work, considering that the masks are highly correlated with the target bitrate and corresponding granularity ratios, **we would first clarify the details of the granularity ratio setting for specific bitrates**. In our method, once the model is trained, we can obtain an index frequency table and assign the codes to each index during the entropy encoding process based on this table. By calculation, for a codebook with 1024 codes, the average code length of all the indices is $L=10.3875$. Supposing an input image with the size of $H\times W$, the amount of its assigned indices range is [$H/16\times W/16$, $H/4\times W/4$], which corresponds to full coarse-grained and fine-grained patch division. For each combination of granularity ratios ($r_1$, $r_2$, $r_3$), we can get its theoretical values of indices and masks by:
>
> $\mathbf{Bpp_{Indices}}=\frac{\frac{H}{4}\times \frac{W}{4}\times r_1+\frac{H}{8}\times \frac{W}{8}\times r_2+\frac{H}{16}\times \frac{W}{16}\times r_3}{(H\times W)}\times L =\frac{(16r_1+4r_2+r_3)\times L}{256}$
>
> $\mathbf{Bpp_{Mask}}=\frac{\frac{H}{4}\times \frac{W}{4}\times r_1+\frac{H}{8}\times \frac{W}{8}\times r_2}{H\times W} =\frac{4r_1+r_2}{256}$
>
> where the total bpp is $(\mathbf{Bpp_{Indices}}+ \mathbf{Bpp_{Mask}}) =\frac{(16L+4)r_1+(4L+1)r_2+r_3}{256}, r_1\geq0, r_2\geq0, r_3\geq0, r_1+r_2+r_3=1$. Consequently, we can generate a query table for the theoretical bpps with ($r_1$, $r_2$, $r_3$).  When receiving a user-given bpp, the model can automatically search the closest theoretical bpp in the query table and assign corresponding granularity ratios. In our investigations, we found that the error between the two values is less than 0.02, and the actual value is always slightly smaller than the theoretical value. Notably, our model allows users to modify the granularity ratios to mitigate such an error and produce better results. We also add the details in our **Appendix A.3 ($\textcolor[RGB]{65, 105, 225}{highlighted\ in\ blue}$)**.
>
> **For the mask setting**, with the given bpp and determined granularity ratios ($r_1$, $r_2$, $r_3$), our model automatically selects the patches with the top $r_3$ percent in order as the coarse ones. Through the encoder $E$, we can get a binary mask $m_3$ (0: not coarse, 1: coarse) and the corresponding elements in $z_3$ that are retained. Subsequently, $E$ removes the coarse-grained patches from the input $I$ based on $m_3$. We repeat the process on the remaining patches at medium granularity to generate $m_2$ and their corresponding elements in $z_2$. Finally, the rest patches are assigned to the fine-grained representation and $m_1$ can be calculated by $1-(m_2)$$\uparrow_2$$-(m_3)$$\uparrow_4$.
>
>
> **Weakness 3**: Typo errors
>
> **Response 3**: Thanks for your careful reading. We have proofread the manuscript and revised the errors. The writing has been improved in the revised manuscript.
>
> **[Ref1]** Matthew Muckley *et al.*, Improving Statistical Fidelity for Neural Image Compression with Implicit Local Likelihood Models.  ICML, 2023.

---

> > ### Comment · Reviewer_bPMe · 2024-11-22
> >
> > Many thanks to the author for addressing my concerns. I have no further questions. I keep my current score unchanged.

---

> > > ### Author Response · Authors · 2024-11-22
> > > **Thanks for your positive approval**
> > >
> > > Dear Reviewer bPMe:
> > >
> > > We are very grateful for your approval of our work, and we believe that your and all reviewers' comments will make our work better. Many thanks again.
> > >
> > > Best regards,
> > >
> > > All the authors

---

### Official Review · Reviewer_WUkA · 2024-10-27

**Soundness:** 3
**Presentation:** 2
**Contribution:** 3
**Rating:** 5
**Confidence:** 4

**Summary:**

The paper Once-for-All: Controllable Generative Image Compression with Dynamic Granularity Adaption introduces Control-GIC, a flexible framework for high-quality image compression that allows fine-grained bitrate control without needing multiple models. Built on a VQGAN foundation, Control-GIC encodes images as variable-length sequences, adapting bitrate based on local image information density, which allows for effective compression adjustments to meet different content complexities. The model includes a granularity-aware encoder that assigns varying levels of detail across image patches, and a probabilistic conditional decoder that reconstructs images with high perceptual quality by aggregating multi-scale features. Experimental results show that Control-GIC outperforms recent state-of-the-art methods in perceptual quality, flexibility, and efficiency, achieving this with a single unified model across a wide bitrate range.

**Strengths:**

1. The authors’ granularity-informed encoder effectively leverages the information density distribution to distill image patches into a hierarchical structure with three levels of granularity. This approach enables compression with a controllable encoding rate, providing a novel solution for adaptable image compression.

2. The paper’s codec framework is lightweight and demonstrates relatively fast encoding and decoding speeds, making it more suitable for practical applications where computational efficiency is a priority.

**Weaknesses:**

1. The experimental results show mixed outcomes compared to existing baselines like HiFiC, without demonstrating a clear advantage.
2. The paper should consider comparing with MS-ILLM [1], which uses a similar VQ-VAE structure, to better position its contributions.
3. Tests on the CLIC 2020 dataset are missing, which would help validate the model's robustness across diverse datasets.
4. Section 3.2 is overly detailed and could be streamlined for clarity and conciseness.
5. Section 3.3 appears to be a direct application of existing methods, which could benefit from additional elaboration or innovation.

[1] Muckley M J, El-Nouby A, Ullrich K, et al. Improving statistical fidelity for neural image compression with implicit local likelihood models. International Conference on Machine Learning. PMLR, 2023: 25426-25443.

**Questions:**

1. How are the masks $m_1$, $m_2$, and $m_3$ obtained? The main text does not seem to provide a clear calculation method for these.
2. If compression is required at a specific bitrate, how should $r_1$, $r_2$, and $r_3$ be determined?

---

> ### Author Response · Authors · 2024-11-21
> **Response to Reviewer WUkA (Part 1)**
>
> ** Weakness 1**: Comparing with HiFiC
>
> ** Response 1**: This is HiFiC trained separately to pursue their best performance on different specific R-D points. In this way, multiple fixed-rate models are necessitated to vary bitrates, leading to dramatic computational costs and inefficient deployment to cater to diverse bitrates and devices. In contrast, our method is customized to achieve flexible bitrate control within one unified model while preserving comparable perceptual quality and compression efficiency, which is the most important in this work. In **Figure 5 (revised manuscript, $\textcolor[RGB]{65, 105, 225}{highlighted\ in\ blue}$)**, we further investigate the model efficiency on four terms: encoding time (s), decoding time (s), BD-rate (BD-LPIPS), and training steps (M). Here, for simple clarity, we provide the BD-LPIPS and the encoding/decoding time on the Kodak dataset in the following table for your reference:
> | Method | Training Steps (M) | BD-rate (BD-LPIPS)↓ | Encoding time (s) | Decoding time (s) |
> |:-----------:|:-------:|:-------:|:-------:|:-------:|
> | HiFiC | 2n | -69.4209 | 0.5306 | 1.4993 |
> | Ours | **0.6** | -72.1545 | **0.0143** | **0.0244** |
>
> For the inference speed, HiFiC suffers from critical time costs in both encoding and decoding. Our method achieves the fastest encoding/decoding time, which is $37\times$ faster than HiFiC in encoding and $67\times$ faster in decoding. Moreover, HiFiC as a single-point training method, requires independent training of $n$ models for $n$ R-D points. The proposed model requires only a single training session that enables compression across various bitrates, with the total training steps being substantially reduced to 0.6 million steps. By comparison, our method can achieve a promising balance among training costs, inference speed, and BD-rate saving.
>
> **Weakness 2**: Comparison with MS-ILLM
>
> **Response 2**: We apologize that the comparison with MS-ILLM is missing. In the revised manuscript, we have compared with MS-ILLM [ref1] on both DIV2K and Kodak datasets, where the results are in **Figure 3 and Figure 4 (revised manuscript, $\textcolor[RGB]{65, 105, 225}{highlighted\ in\ blue}$)**, respectively. As we can see, MS-ILLM is trained for specific R-D points, which achieves relatively higher quantitative performance in LPIPS and DISTS. However, even though, our method exhibits comparable performance in FID and KID, and better NIQE than MS-ILLM. Here, for simple clarity, we provide training steps (M), BD-rate (BD-LPIPS) results on the DIV2K dataset, and encoding/decoding time (s) on the Kodak dataset for quick comparison using VVC as an anchor, as follows:
> | Method | Training Steps (M) | BD-rate (BD-LPIPS)↓ | Encoding time (s) | Decoding time (s) |
> |:-------:|:-------:|:-------:|:-------:|:-------:|
> | BPG | - | 34.5704 | 1.1001 | 0.3514 |
> | M&S | 2n | 33.4252 | 0.0923 | 0.0514 |
> | SCR | 9.4 | 38.9885 | 0.7924 | 0.6477 |
> | CTC | 3 | -7.8118 | 3.5067 | 2.0880 |
> | HiFiC | 2n | -72.6776 | 0.5306 | 1.4993 |
> | CDC | 1.5n |-48.2543 | 0.1069 | 1.1054 |
> | MS-ILLM | 2n |-74.0735 | 0.1094 | 0.0713 |
> | Ours | **0.6** | -67.5585 | **0.0143** | **0.0244** |
>
> Our method shows the fastest encoding/decoding time, which is $7\times$ faster than MS-ILLM in encoding, and $3\times$ faster than MS-ILLM in decoding. The comparisons demonstrate that our method can achieve a better trade-off between efficiency and effectiveness while preserving high flexibility to support variable bitrates in one model.
>
> **Weakness 3**: Tests on the CLIC 2020 dataset
>
> **Response 3**: Thanks for your suggestion. We have provided the comparison on the CLIC2020 dataset, where the R-D performance and qualitative performance are shown in **Figure 10 (Appendix, $\textcolor[RGB]{65, 105, 225}{highlighted\ in\ blue}$)**. Here, we provide the comparison in the following table for your reference:
> | Method | Bpp↓ | LPIPS↓ | DISTS↓ | FID↓ | KID↓ | PSNR↑ | NIQE↓|
> |:-----------:|:-------:|:-------:|:-------:|:-------:|:-------:|:-------:|:-------:|
> | M&S | 0.2921 | 0.1144 | 0.1270 | 33.2748| 0.0164 | 35.0479 | 5.0712 |
> | CTC | 0.2714 | 0.0961 | 0.1100 | 24.4000| 0.0127 | 35.6997 | 4.6345 |
> | HiFiC | 0.2244 | 0.0331 | 0.0532 | 3.4804 | 0.0007 | 31.5913 | 3.8988 |
> | CDC | 0.2284 | 0.0548 | 0.0411| 2.8103 | 0.0006 | 28.7806 | 4.4357 |
> | Ours | 0.2243 | 0.0342 | 0.0503| 3.3644 | 0.0007 | 28.9413 | 3.4638 |
>
> It can be seen that **Control-GIC** achieves superior performance in most metrics over M&S and variable-rate method CTC. Compared to generative compression methods which are trained separately for multiple R-D points, our **Control-GIC** still maintains competitive performance, which validates that our method can achieve optimal trade-off between flexibility and effectiveness.
>
> **[Ref1]** Matthew Muckley *et al.*, Improving Statistical Fidelity for Neural Image Compression with Implicit Local Likelihood Models.
>  ICML, 2023.

---

> ### Author Response · Authors · 2024-11-21
> **Response to Reviewer WUkA (Part 2)**
>
> **Weakness 4 & 5**: Writing improvement in Section 3.2 and Section 3.3.
>
> **Response 4 & 5**: Thanks for your suggestion, we have re-organized the description and explanation in Section 3.2 and Section 3.3, which are **$\textcolor[RGB]{147, 112, 219}{highlighted\ in \ purple}$ in the revised manuscript**.
>
> **Questions 1 & 2**: Clarity on the masks $m_1$, $m_2$, $m_3$ acquisition and setting of $r_1$, $r_2$, $r_3$
>
> **Response 1 & 2**: In this work, considering that the masks are highly correlated with the target bitrate and corresponding granularity ratios, **we would first clarify the details of the granularity ratio setting for specific bitrates**. In our method, once the model is trained, we can obtain an index frequency table and assign the codes to each index during the entropy encoding process based on this table. By calculation, for a codebook with 1024 codes, the average code length of all the indices is $L=10.3875$. Supposing an input image with the size of $H\times W$, the amount of its assigned indices range is [$H/16\times W/16$, $H/4\times W/4$], which corresponds to full coarse-grained and fine-grained patch division. For each combination of granularity ratios ($r_1$, $r_2$, $r_3$), we can get its theoretical values of indices and masks by:
>
> $\mathbf{Bpp_{Indices}}=\frac{\frac{H}{4}\times \frac{W}{4}\times r_1+\frac{H}{8}\times \frac{W}{8}\times r_2+\frac{H}{16}\times \frac{W}{16}\times r_3}{(H\times W)}\times L =\frac{(16r_1+4r_2+r_3)\times L}{256}$
>
> $\mathbf{Bpp_{Mask}}=\frac{\frac{H}{4}\times \frac{W}{4}\times r_1+\frac{H}{8}\times \frac{W}{8}\times r_2}{H\times W} =\frac{4r_1+r_2}{256}$
>
> where the total bpp is $(\mathbf{Bpp_{Indices}}+ \mathbf{Bpp_{Mask}}) =\frac{(16L+4)r_1+(4L+1)r_2+r_3}{256}, r_1\geq0, r_2\geq0, r_3\geq0, r_1+r_2+r_3=1$. Consequently, we can generate a query table for the theoretical bpps with ($r_1$, $r_2$, $r_3$). When receiving a user-given bpp, the model can automatically search the closest theoretical bpp in the query table and assign corresponding granularity ratios. In our investigations, we found that the error between the two values is less than 0.02, and the actual value is always slightly smaller than the theoretical value. Notably, our model allows users to modify the granularity ratios to mitigate such an error and produce better results. We also add the details in our **Appendix A.3 ($\textcolor[RGB]{65, 105, 225}{highlighted\ in\ blue}$)**.
>
> **For the mask setting**, with the given bpp and determined granularity ratios ($r_1$, $r_2$, $r_3$), our model automatically selects the patches with the top $r_3$ percent in order as the coarse ones. Through the encoder $E$, we can get a binary mask $m_3$ (0: not coarse, 1: coarse) and the corresponding elements in $z_3$ that are retained. Subsequently, $E$ removes the coarse-grained patches from $I$ based on $m_3$. We repeat the process on the remaining patches at medium granularity to generate $m_2$ and their corresponding elements in $z_2$. Finally, the rest patches are assigned to the fine-grained representation and $m_1$ can be calculated by $1-(m_2)$$\uparrow_2$$-(m_3)$$\uparrow_4$.

---

> ### Author Response · Authors · 2024-11-24
> **Request for your feedback**
>
> Dear Reviewer WUkA:
>
> Hoping this message finds you well. Your comments and the provided insights are very valuable to our work. We will attach these analysis and additional experiments to our final version. As the rebuttal-discussion period is nearing its end, could you please review our response to see if it addresses your concerns? Your timely feedback will be extremely valuable to us. Could you read and let us know if there are more questions? We would be very grateful! Your decision is of utmost importance to us, and we earnestly hope that you will consider the significant contribution of this research to the field of image compression. Thank you very much!
>
> Best regards,
>
> All the authors

---

> > ### Comment · Reviewer_WUkA · 2024-11-26
> >
> > 1. The proposed method shows a noticeable performance gap in compression compared to MS-ILLM, which has a similar VQ-VAE structure.
> > 2. On the Kodak dataset, the PSNR metric is relatively low at higher bpp values.
> > 3. How is the query table for the theoretical bpps with $(r_1, r_2, r_3)$ generated, and does it generalize well to different datasets?

---

> > > ### Author Response · Authors · 2024-12-02
> > > **Response to Reviewer WUkA (Part 1)**
> > >
> > > Thanks for your valuable comments and timely feedback. We hope this response can address your concerns. Considering that the manuscript PDF has closed for revisions, we guarantee that all the results will be updated in our final version.
> > >
> > > **Comment 1**: Comparison to MS-ILLM.
> > >
> > > **Response 1**: We fully understand your concerns about the performance gap between MS-ILLM and our Control-GIC. Before the discussion and comparisons, we would like to clarify that, MS-ILLM just applies a VQVAE-based adversarial discriminator to optimize likelihood functions. It does not employ VQVAE architecture as its backbone as other VQGAN methods **[Ref1-3]**. Instead, it adheres to the autoencoder (AE) framework that is consistent with HiFiC, as referenced in its supplementary material of the public paper under **Section A.1 Autoencoder Pretraining**: “For pretraining the autoencoder, we used the same overall approach as HiFiC”. MS-ILLM introduces the VQ-VAE framework within the context of labeler pretraining, as detailed in **Section 4.3 Choice of labeling function** and **Section A.2 Labeler Pretraining** (its public paper). Upon the completion of this training phase, the VQ-VAE component is subsequently discarded. The primary function of incorporating VQ-VAE in this study is to enhance the discriminative capacity of the labeler by transforming the binary classification of patches—originally labeled as either true or false—into 1024-dimensional (same as the codebook size of VQ-VAE) representation.
> > >
> > > Consequently, the underlying motivations and structural frameworks of our Control-GIC and MS-ILLM are fundamentally distinct. As you commented in **Strength**, the key contribution of our method is the flexible compression with controllable bitrates, achieved within a single model, while simultaneously maintaining comparable reconstruction capability and superior codec speed. Our network design is to strike a balance among compression efficacy, effectiveness, and model flexibility. To provide a more intuitive comparison, we also conduct experiments on single R-D points as MS-ILLSM, where the results for all generative methods on Kodak and DIV2K datasets are accessible at the provided anonymous link:
> > > **https://anonymous.4open.science/r/F11E/comparison.pdf**. Here, we also report some quantitative performance on several bpp in the following table for your reference. Notably, since there are merely R-D points of CDC that are consistent with other methods and ours. In this table, its quantitative performance is not included.
> > >
> > > Kodak：
> > >
> > >
> > > | Method | Bpp↓ | LPIPS↓ | DISTS↓ | NIQE↓ | Enc. (s) | Dec. (s) |
> > > |:------:|:----:|:------:|:------:|:-----:|:--------:|:--------:|
> > > | HiFiC  | 0.1491 | 0.0704 | 0.1037 | 3.4840 | 0.5346 | 1.4923 |
> > > | MS-ILLM | 0.1535 | 0.0546 | 0.0810 | 3.2693 | 0.1065 | 0.0724 |
> > > | Ours   | 0.1388 | 0.0531 | 0.0809 | 2.9892 | 0.0146 | 0.0236 |
> > > |        |       |        |        |       |          |          |
> > > | HiFiC  | 0.3032 | 0.0466 | 0.0817 | 3.4096 | 0.5335 | 1.4968 |
> > > | MS-ILLM | 0.2961 | 0.0323 | 0.0613 | 3.3430 | 0.1026 | 0.0756 |
> > > | Ours   | 0.2856 | 0.0356 | 0.0698 | 2.9230 | 0.0142 | 0.0246 |
> > >
> > > DIV2K：
> > >
> > > | Method | Bpp↓ | LPIPS↓ | DISTS↓ | NIQE↓ |
> > > |:-------:|:-----:|:------:|:------:|:-----:|
> > > | HiFiC   | 0.1458 | 0.0620 | 0.0762 | 3.4286 |
> > > | MS-ILLM | 0.1493 | 0.0646 | 0.0556 | 3.1949 |
> > > | Ours    | 0.1466 | 0.0528 | 0.0655 | 2.9994 |
> > > |        |       |        |        |       |
> > > | HiFiC   | 0.2909 | 0.0411 | 0.0576 | 3.3760 |
> > > | MS-ILLM | 0.2801 | 0.0403 | 0.0405 | 3.2146 |
> > > | Ours    | 0.2822 | 0.0355 | 0.0460 | 2.9592 |
> > >
> > > One can see that our method outperforms existing SOTA methods in almost all three metrics (LPIPS, DISTS, and NIQE) across both datasets at low bit rates. While at higher bit rates, our method can also perform comparably. Moreover, it can also be observed that our method achieves faster encoding/decoding time compared to HiFiC and MS-ILLM (for the full comparison see **Figure 5: revised manuscript**, and **Response 2: Former Response Part 1**). These comparisons demonstrate that our method can achieve a better trade-off between efficiency and effectiveness.
> > >
> > >
> > > **[Ref1]** Qi Mao, et al, Extreme image compression using fine-tuned vqgan models.DCC, 2024.
> > >
> > > **[Ref2]** Naifu Xue, Qi Mao, et al, Unifying Generation and Compression: Ultra-low bitrate Image Coding Via Multi-Stage Transformer. ICME.2024.
> > >
> > > **[Ref3]** Zhaoyang Jia et al. Generative Latent Coding for Ultra-Low Bitrate Image Compression, CVPR, 2024.

---

> > > ### Author Response · Authors · 2024-12-02
> > > **Response to Reviewer WUkA (Part 2)**
> > >
> > > **Comment 2**: The relatively low PSNR at a higher bitrate.
> > >
> > > **Response 2**: Thank you for pointing out this issue, which is also of important concern to us. We have investigated the underlying reasons for the PSNR differences and observed that other methods, such as HiFiC and MS-ILLM, incorporate entropy models to more precisely capture the latent variables and the generator with additional guidance information. The latent variables and additional data become increasingly accurate with higher bitrates, thereby boosting PSNR. Nonetheless, this approach incurs a computational burden and complicates the controllability of bitrate, which does not actually meet the high flexibility requirements of our method. Therefore, in our experiments, besides some quantitative metrics including PSNR, we may pay more attention to the bitrate control and generative ability. In the qualitative comparison (**Figure 6, revised manuscript**), it can be observed that our method outperforms existing methods in perception while others prioritizing PSNR show marginal reconstruction improvements and are weak in flexible bitrate and encoding/decoding speed.
> > >
> > > **Comment 3**: The generation of the query table.
> > >
> > > **Response 3**: We have explored the optimal granularity ratio $(r_1, r_2, r_3)$ corresponding to target bpps. We consider the best option is that given the target bpp value, the bpp output by the network needs to be equal to the target bpp, and the combination of granularities ensures the highest image quality. In **Response 1 & 2 in the former Response Part 2**, we have explained the mapping relationship between $(r_1, r_2, r_3)$ and the target bpp as follows:
> > >
> > > $Bpp=f( r1, r2, r3)= \frac{(16L+4)r_1+(4L+1)r_2+r_3}{256}, r_1, r_2, r_3\geq0, r_1+r_2+r_3=1$,
> > >
> > > Intuitively, the patches with different granularities significantly affect the quality of the compressed image. For example, the finest granularity leads to higher image quality, and the coarsest granularity leads to lower image quality. Let $(q_1, q_2, q_3)$ denote the reconstruction quality of each granularity $(r_1, r_2, r_3)$, respectively. Since an image is a patchwork of regions represented by different granularities, the overall quality of the image can be approximated as a weighted sum of the quality of each granularity:
> > >
> > > $Q=g( r_1, r_2, r_3) = r_1 \cdot q_1+r_2\cdot q_2+r_3\cdot q_3$ ,
> > >
> > > Therefore, our goal is to maximize the image quality $Q$ while satisfying the bpp constraints $Bpp$. Note that $Q$ is an indicator of image quality and does not directly equate to the final quality. Specifically, we use LPIPS as a metric and compress the Kodak dataset using full-fine ($r_1$=100%), full-medium ($r_2$=100%), and full-coarse ($r_3$=100%) to obtain the LPIPS values as image quality under three conditions, *i.e.*, $q_1=0.024820354$, $q_2=0.065266579$, $q_3=0.273727685$ and the objective becomes to minimize:
> > >
> > > $Q = g(r_1,r_2,r_3) = 0.024820354\cdot r1+0.065266579\cdot r2+0.273727685\cdot r3$,
> > >
> > > We employ the BFGS (Broyden-Fletcher-Goldfarb-Shanno) optimization algorithm to solve for $(r_1, r_2, r_3)$ and add several bpps and the corresponding $(r_1, r_2, r_3)$ to our query table.
> > >
> > > We also evaluate the generalizability of the query table on different datasets, including Kodak, DIV2K, and CLIC2020. We assign the optimal granularity ratio to the six target bpp values in the interval of 0.1~0.6 and compress the three datasets to obtain the actual bpp. Finally, we can have an average error of **no more than 0.022bpp** on the three datasets (as shown in the following table), which proves the generalizability of the query table.
> > >
> > > | Target Bpp | $(r_1, r_2, r_3)$ | Kodak | DIV2K | CLIC2020 |
> > > |:---:|:-------:|:-------:|:-------:|:-------:|
> > > | 0.1 | (0, 60%, 40%) | 0.13030158 | 0.13025940 | 0.12625429 |
> > > | 0.2 | (13%, 67%, 20%) | 0.20670658 | 0.20766181 | 0.20123426 |
> > > | 0.3 | (33%, 49%, 18%) | 0.30056593 | 0.30191789 | 0.29273910 |
> > > | 0.4 | (51%, 42%, 7%) | 0.39156765 | 0.39256203 | 0.38125384 |
> > > | 0.5 | (70%, 30%, 0) | 0.48094347 | 0.4807709 | 0.46908913 |
> > > | 0.6 | (90%, 10%, 0) | 0.56892310 | 0.56806642 | 0.55578612 |
> > > | Avg Diff  |-| 0.01602331 | 0.01640661 | 0.02143672 |

---

> ### Comment · Reviewer_WUkA · 2024-12-02
>
> 1. Considering performance, I believe it is acceptable for a variable bitrate solution to slightly underperform compared to a fixed bitrate solution. Similarly, if we aim for a lightweight approach, achieving faster encoding and decoding speeds with comparable performance is also acceptable. However, among the baselines compared in this paper, I did not observe such a case.
>
> 2. It seems that the paper does not specify how the query table is determined. Additionally, you mentioned using BFGS to determine $(r_1, r_2, r_3)$. Why was this method chosen? What is its principle? Can it guarantee an optimal result? The authors appear not to have provided an explanation.
>
> Therefore, I maintain my score.

---

> ### Author Response · Authors · 2024-12-02
> **Response to Reviewer WUkA**
>
> We much appreciate your timely feedback in your busy schedule. We are also very glad that you can provide such in-depth comments and analysis.
>
> **Comment 1**: Model performance.
>
> **Response 1**: In this work, our goal is to design a flexible, efficient, and effective generative compression method that accommodates a wide spectral range of bitrates in a unified model. Therefore, we compare with existing methods in 3 aspects:
>
> **1)** Variable-rate methods: SCR and CTC. By the comparison in **Figure 3, 4, and 5 (revised manuscript)**, our method achieves superior quantitative performance in most metrics and faster compression speed.
>
> **2)** Generative methods: HiFiC, CDC, MS-ILLM, and MRIC. As you pointed out, their R-D performance is sometimes better than ours. But in perceptual quality LPIPS or non-reference quality assessment, our method actually achieves comparable performance while keeping faster speed in a variable-rate framework. Besides, in **Figure 6 (revised manuscript)**, the results indicate that our method can reconstruct more preferable textures. Furthermore, based on the above response, it can be seen that even in a single R-D point pattern, our method is still comparable or superior with less encoding/decoding time.
>
> **3)** Traditional CNN-based or non-deep learning-based codecs M&S, BPG, and VVC. These methods are prone to reconstruct compressed images based on the pixel-wise constraint (*i.e.* MSE), which shows obviously lower perceptual performance and worse subjective quality than ours as well as other generative compression methods.
>
> Based on the above, we think that our method can achieve a better balance between flexibility, performance, and efficiency than existing methods. Therefore, we sincerely hope that you can reconsider the contributions and effects of this work.
>
> **Comment 2**: Query table.
>
> **Response 2**: The BFGS algorithm is widely recognized for its efficacy in addressing unconstrained optimization problems **[Ref1-3]**. Its core mechanism involves iteratively refining the search direction by approximating the inverse Hessian, which is the matrix of second derivatives of the objective function. Based on the findings in existing research **[Ref1]**, BFGS has been demonstrated to benefit the convergence speed and model performance, which are essential for the efficient construction of our query table. Moreover, our model has demonstrated robustness across various granularity combinations. For clarity, here we provide some validation results on the Kodak dataset as follows:
>
> | | Granularity Ratio | | | Bpp | LPIPS |
> |:---:|:---:|:---:|:---:|:---:|:---:|
> | $r_1$ | $r_1$ | $r_3$ | | | |
> | 40% | 55% | 5% | | 0.3503 | 0.0304 |
> | 40% | 50% | 10% | | 0.3500 | 0.0310 |
> | 43% | 42% | 15% | | 0.3490 | 0.0312 |
>
> We can observe that fine-tuning the ratio of the three granularities within a given combination has a minimal impact on image quality. The results indicate that varying the granularity ratios yields closely aligned bpp outcomes and further suggests that a certain target bpp can correspond to multiple solutions. Notably, there is no significant variation in the final image quality. This stability can be attributed to the model's comprehensive learning across all three granularities, which endows it with considerable robustness.
>
> Based on these two reasons, we have selected the BFGS algorithm as our optimization method. Since the manuscript PDF revision has concluded, we guarantee that the pertinent algorithmic principles and processes will be detailed in the final version.
>
> Last, we would like to thank you for your active discussion and valuable comments. We would be very grateful if you could kindly reconsider our contributions and explanations.
>
> **[Ref1]** Nocedal, Jorge, and Stephen J. Wright, eds. Numerical optimization. New York, NY: Springer New York, 1999.
>
> **[Ref2]** Gill, Philip E., Walter Murray, and Margaret H. Wright. Practical optimization. Society for Industrial and Applied Mathematics, 2019.
>
> **[Ref3]** Bottou, Léon, Frank E. Curtis, and Jorge Nocedal. "Optimization methods for large-scale machine learning." SIAM review 60, no. 2 (2018): 223-311.

---

### Official Review · Reviewer_P96B · 2024-11-02

**Soundness:** 1
**Presentation:** 1
**Contribution:** 1
**Rating:** 3
**Confidence:** 5

**Summary:**

This paper targets at addressing the challenge of rate adaption proplem for generative image compression and proposes a controllable generative image compression framework. The paper represents the image patches of sequential spatially variant VQ-indices to support precise variable rate control and adaption. A non-parametric statistical entropy coding is devised to encode the VQ-indices losslessly.
A probabilistic conditional decoder is proposed to aggregate historic encoded multi-granularity representations, achieving realism improvements.

**Strengths:**

The paper proposes to flexibly determine a proper allocation of granularity for patches, supporting dynamic adjustment for VQ-indices and make the framework capable of fine-grained bitrate adaptation.

**Weaknesses:**

- In Figure 3, the proposed method shows worse performance than CDC for DISTS, and is also worse than CDC at the high bitrate range. Could you give some analysis?

- In Figure 4, why do you compare only with BPG, CTC, and HiFiC, instead of aligning with the methods used in Figure 3?

- In Figure 3 and Figure 4, most metrics are nearly reaching saturation when the bpp increases. Is there obvious difference for the visualization quality when the bpp increases at the high bitrate range? For example, in Figure 6, when the bpp increases, I do not see obvious enhancement between r2=40%(bpp=0.3864) and r2=32.9%(0.4171).

- Visualization analysis about the influence of the granularity of image patches. Will increasing the image patch size lead to block artifacts in the reconstructed images? And can making the image patch size smaller solve some artifacts in VQ based methods, such as the unsatisfactory artifacts for small faces?

**Questions:**

see the weakness part

---

> ### Author Response · Authors · 2024-11-21
> **Response to Reviewer P96B (Part 1)**
>
> **Weakness 1**: Compare with CDC in DISTS
>
> **Response 1**: As we know, DISTS measures the structure and texture similarity between compressed and uncompressed images. CDC is implemented on the conditional diffusion model which leverages encoded latent variables containing sufficient structural and semantical information as the condition. Besides, a high bitrate indicates more accurate conditional information can be exploited to facilitate the reconstruction in CDC, resulting in better DISTS scores. We would like to explain that, in generative compression, the LPIPS, FID, or NIQE are also important metrics. As shown in **Figure 3 and Figure 4 (revised manuscript, $\textcolor[RGB]{65, 105, 225}{highlighted\ in\ blue}$)**, our method achieves better performance.
>
> This can be attributed to our dynamic granularity adjustment approach and the generative ability of VQGAN. Moreover, in literature, the methods including CDC trained separately to pursue their best performance on different specific R-D points often exhibit superiority over variable-rate compression methods. Our method is customized to achieve flexible bitrate control within one unified model while preserving comparable perceptual quality and compression efficiency. Even though, our method still performs comparably in most metrics across multiple datasets against CDC.
>
> In addition, for the inference speed, CDC applies a lightweight diffusion variational autoencoder which benefits the encoding process but struggles with more decoding time due to its iterative reverse process. Our method achieves $1.5\times$ faster than CDC in encoding, and $45\times$ faster in decoding. The comparison on the Kodak dataset is as follows:
> | Method | Training Steps (M) | BD-rate (BD-LPIPS)↓ | Encoding time (s) | Decoding time (s) |
> |:-----------:|:-------:|:-------:|:-------:|:-------:|
> | CDC | 1.5n | -39.3307 | 0.1069 | 1.1054 |
> | Ours | 0.6 | -72.1545 | 0.0143 | 0.0244 |

---

> ### Author Response · Authors · 2024-11-21
> **Response to Reviewer P96B (Part 2)**
>
> **Weakness 2**: The comparison in Figure 4
>
> **Response 2**: We apologize that some results are missing in Figure 3. This is because our initial view posited that the analysis of the Kodak dataset would suffice to ascertain the efficacy of all the methods. Therefore, in Figure 4, on the DIV2K dataset, we select the variable-rate method CTC and GAN-based method according to the performance on the Kodak, which are related to ours (variable-rate and GAN-based). The BPG as a classical method on specific compression rates is applied as a reference. Besides, to provide more comprehensive comparisons, we introduce another two metrics FID and KID to help measure their performance. We appreciate your comments that the complete comparisons on DIV2K are more important than those on Kodak actually. In the revised manuscript, we have included the additional comparisons in Figure 4 (revised manuscript, $\textcolor[RGB]{65, 105, 225}{highlighted\ in\ blue}$). Our Control-GIC surpasses almost all the methods across most distinct metrics and exhibits a significant improvement in NIQE. We also include more generative models (MRIC **[Ref1]** and MS-ILLM **[Ref2]**) in Figure 4. As we can see, these two methods are trained for specific R-D points, which achieve relatively higher quantitative performance in LPIPS and DISTS. However, even though, our method exhibits comparable performance in FID and KID, and better NIQE than these two most state-of-the-art methods. Here, for simple clarity, we provide training steps (M), BD-rate (BD-LPIPS) results on the DIV2K dataset, and encoding/decoding time (s) on the Kodak dataset for quick comparison using VVC as an anchor, as follows:
> | Method | Training Steps (M) | BD-rate (BD-LPIPS)↓ | Encoding time (s) | Decoding time (s) |
> |:-------:|:-------:|:-------:|:-------:|:-------:|
> | BPG | - | 34.5704 | 1.1001 | 0.3514 |
> | M&S | 2n | 33.4252 | 0.0923 | 0.0514 |
> | SCR | 9.4 | 38.9885 | 0.7924 | 0.6477 |
> | CTC | 3 | -7.8118 | 3.5067 | 2.0880 |
> | HiFiC | 2n | -72.6776 | 0.5306 | 1.4993 |
> | CDC | 1.5n |-48.2543 | 0.1069 | 1.1054 |
> | MRIC | 3n |-81.9656 | 0.0598 | 0.0352 |
> | MS-ILLM | 2n |-74.0735 | 0.1094 | 0.0713 |
> | Ours | **0.6** | -67.5585 | **0.0143** | **0.0244** |
>
> It can be seen that the variable-rate approach represented by SCR does not perform as well as models trained on a single point (*e.g.*, M&S), while CTC obtains superior performance with a larger number of parameters (399M, computed from the official open-source code), but with a very large drop in model efficiency (the encoding time is $38\times$ times longer and the decoding time is $40\times$ times longer than M&S). MRIC and MS-ILLM trained separately for each compression rate obtained very close BD-rate savings with our **Control-GIC**, outperforming other methods. Our method achieves the fastest encoding/decoding time, which is $7\times$ faster than MS-ILLM and $4\times$ faster than MRIC in encoding, and $3\times$ faster than MS-ILLM and $1.5\times$ faster than MRIC in decoding. Moreover, the single-point training methods (*e.g.*, HiFiC, CDC, MRIC, MS-ILLM) require independent training of $n$ models for $n$ R-D points. Our proposed model requires only a single training session that enables compression across various bitrates, with the total training steps being substantially reduced to 0.6 million steps. By comparison, our method can achieve a promising balance among training costs, inference speed, and BD-rate saving.
>
> **Weakness 3**: Performance saturation when the bpp increases
>
> **Response 3**: As you pointed out, in most cases, when bpp increases to a certain extent, the improvement in objective metrics gradually becomes very marginal. Besides, the difference in reconstructed images by high bitrates in human visual perception also decreases to very low. To more intuitively illustrate such differences, in **Figure 7 (revised manuscript, $\textcolor[RGB]{0, 205, 205}{highlighted\ in\ cyan}$)**, we calculate the difference maps between compressed images and original one, one can see that, as the bpp increases, there is also a decrease in information loss, which indicates an improved capability of the model to for better image details recovery.
>
> **[Ref1]** Eirikur Agustsson *et al.*, Multi-realism image compression with a conditional generator. CVPR, 2023.
>
> **[Ref2]** Matthew Muckley *et al.*, Improving Statistical Fidelity for Neural Image Compression with Implicit Local Likelihood Models. ICML, 2023.

---

> ### Author Response · Authors · 2024-11-21
> **Response to Reviewer P96B (Part 3)**
>
> **Weakness 4**: Performance different patch size
>
> **Response 4**: When the patch size is increased, then a single code needs to represent more image regions, and texture distortion occurs when local region details and textures are complex. Reducing the patch size can effectively solve the block artifact problem in the VQ method, and the reconstruction of local small faces in the image will be improved. In this work, we change the patch size based on the assignment of granularity ratios, where the results are shown in **Figure 9 (Appendix A. 4, $\textcolor[RGB]{0, 205, 205}{highlighted\ in\ cyan}$)**. We use the results of CDC and MS-ILLM are used as references. The small face poses a critical challenge in image compression. Both CDC and MS-ILLM struggle to obtain promising generations. When we assign overly high coarse- and medium-granularity ratios on the face area, *i.e.* large patch size, the models also cannot recover the face details well. When we set the patches in the face region as fine-granularity ones and reduce their patch size, the artifacts can be further alleviated, where the generated images reveal clearer details. Consequently, our model using (40%, 40%, 20%) yields a better reconstruction and lower bpp. Despite the superiority, due to the complexity of small faces, there is still a need for more targeted design.

---

> ### Author Response · Authors · 2024-11-22
> **Request for Clarification on Score Reduction**
>
> Dear Reviewer,
>
> Thank you for your feedback. We noticed the lower score and would like to understand the reasons behind it. Could you please share any specific concerns or areas for improvement? We are committed to addressing them to enhance our work.
>
> Best regards,
>
> All the authors

---

> ### Author Response · Authors · 2024-11-24
> **Request for your feedback**
>
> Dear Reviewer P96B:
>
> Hoping this message finds you well. Your comments and the provided insights are very valuable to our work. Besides previous responses and revisions, we now add more experiments and analyses, including 1) well-developed theoretical support and detailed descriptions of the granularity assignments; 2) results of additional datasets; and 3) exploration of the extremely low bitrates. We will attach these analyses and additional experiments to our final version. As the rebuttal-discussion period is nearing its end, could you please review our response to see if it addresses your concerns? Your timely feedback will be extremely valuable to us. Could you read and let us know if there are more questions? We would much appreciate the clarification related to soundness, presentation, and contribution. Your decision is of utmost importance to us, and we earnestly hope that you will consider the significant contribution of this research to the field of image compression. Thank you very much!
>
> Best regards,
>
> All the authors

---

### Official Review · Reviewer_qmzM · 2024-11-03

**Soundness:** 3
**Presentation:** 3
**Contribution:** 3
**Rating:** 8
**Confidence:** 5

**Summary:**

This paper introduces Control-GIC, a controllable generative image compression framework aimed at addressing the limitations of existing generative image compression methods in achieving flexible bitrate adjustment. Built upon the VQGAN framework, Control-GIC incorporates multi-granularity encoding mechanisms and a probabilistic conditional decoder to achieve flexible bitrate control and high-quality image reconstruction. Both qualitative and quantitative results demonstrate the effectiveness of the proposed method.

**Strengths:**

1. The proposed Control-GIC introduces a unified model that allows dynamic bitrate adjustment, which effectively solves the inefficiencies faced by existing models that need multiple fixed-rate versions.
2. The granularity-informed encoder and probabilistic conditional decoder are well-designed to achieve efficient encoding and high perceptual fidelity.
3. Experimental results show superior performance over state-of-the-art methods, demonstrating both flexibility and effectiveness in compression.

**Weaknesses:**

1. The paper lacks sufficient details on how the features are divided into different granularities in Section 3.1.
2. The DIV2K comparisons in Figure 4 do not include evaluations against important baselines like VVC, M&S, and other methods (presented in Figure 3), which limits the completeness of the analysis.
3. The paper does not compare Control-GIC with other VQ-based methods, such as GLC [1], Mao et al. [2], and UIGC [3], which would provide a better context for understanding the model's relative performance.
4. It is suggested that the authors add more comparisons with results from other datasets featuring different image sizes to enhance the robustness, such as CLIC-dataset.


References:
[1] Jia Z, Li J, Li B, et al. Generative Latent Coding for Ultra-Low Bitrate Image Compression[C]//Proceedings of the IEEE/CVF Conference on Computer Vision and Pattern Recognition. 2024: 26088-26098.
[2] Qi Mao, et al, Extreme image compression using fine-tuned vqgan models. DCC, 2024.
[3] Naifu Xue, Qi Mao, et al, Unifying Generation and Compression: Ultra-low bitrate Image Coding Via Multi-stage Transformer.ICME.2024.

**Questions:**

1. Why is the mask ratio set to 50%, 40%, 10% for fine, medium, and coarse features in the training setting? Please provide further justification for this choice.
2. In Mao et al.'s papers [2, 3], the bitrate used in VQGAN is significantly lower, ranging from <0.05 bpp to ≤0.03 bpp, while in GIC, the bpp ranges from 0.1 to 0.6 bpp. Could you explain why the bitrate range cannot be lower in Control-GIC, and if it is possible to extend the model to support lower bitrates?



References:
[1] Qi Mao, et al, Extreme image compression using fine-tuned vqgan models.DCC, 2024.
[2] Naifu Xue, Qi Mao, et al, Unifying Generation and Compression: Ultra-low bitrate Image Coding Via Multi-stage Transformer.ICME.2024.

---

> ### Author Response · Authors · 2024-11-21
> **Response to Reviewer qmzM (Part 1)**
>
> **Weakness 1**: Clarify the details on the granularity division in Section 3.1
>
> **Response 1**: We are sorry that some details are missing, which makes you confused. As shown in **Figure 2 (revised manuscript)**, given an image $x$ divided into serial patches, the encoder $E$ first calculates the entropy values of all patches and sorts them from low to high. We define three types of granularities (coarse, medium, high) for these patches. Once model trained, we can obtain an index frequency table and assign the codes to each index during the entropy encoding process based on this table. By calculation, for a codebook with 1024 codes, the average code length of all the indices is $L=10.3875$. Supposing an input image with the size of $H\times W$, the amount of its assigned indices range is [$H/16\times W/16$, $H/4\times W/4$], which corresponds to full coarse-grained and fine-grained patch division. For each combination of granularity ratios ($r_1$, $r_2$, $r_3$), we can get its theoretical values of indices and masks by:
>
> $\mathbf{Bpp_{Indices}}=\frac{\frac{H}{4}\times \frac{W}{4}\times r_1+\frac{H}{8}\times \frac{W}{8}\times r_2+\frac{H}{16}\times \frac{W}{16}\times r_3}{(H\times W)}\times L =\frac{(16r_1+4r_2+r_3)\times L}{256}$
>
> $\mathbf{Bpp_{Mask}}=\frac{\frac{H}{4}\times \frac{W}{4}\times r_1+\frac{H}{8}\times \frac{W}{8}\times r_2}{H\times W} =\frac{4r_1+r_2}{256}$
>
> where the total bpp is $(\mathbf{Bpp_{Indices}}+ \mathbf{Bpp_{Mask}}) =\frac{(16L+4)r_1+(4L+1)r_2+r_3}{256}, r_1\geq0, r_2\geq0, r_3\geq0, r_1+r_2+r_3=1$. Consequently, we can generate a query table for the theoretical bpps with ($r_1$, $r_2$, $r_3$). When receiving a user-given bpp, the model can automatically search the closest theoretical bpp in the query table and assign corresponding granularity ratios. In our investigations, we found that the error between the two values is less than 0.02, and the actual value is always slightly smaller than the theoretical value. Notably, our model allows user to modify the granularity ratios to mitigate such an error and produce better results. Here, we provide a simplified query table that describes the correlations between theoretical bpp and granularity ratios, as an example. Our model enables a very meticulous control of bitrates. We also add the details in our **Appendix A.3 ($\textcolor[RGB]{65, 105, 225}{highlighted\ in\ blue}$)**.
> | | Granularity Ratio | | | Bpp |
> |:---:|:---:|:---:|:---:|:---:|
> | $r_1$ | $r_1$ | $r_3$ | | |
> | 0 | 23% | 77% | | 0.070 |
> | 10% | 67% | 23% | | 0.187 |
> | 37% | 46% | 17% | | 0.330 |
> | 61% | 30% | 9% | | 0.460 |
> | 90% | 10% | 0 | | 0.616 |

---

> > ### Author Response · Authors · 2024-11-24
> > **Request for your feedback**
> >
> > Dear Reviewer qmzM:
> >
> > Hoping this message finds you well. Your comments and the provided insights are very valuable to our work. We will attach these analysis and additional experiments to our final version. As the rebuttal-discussion period is nearing its end, could you please review our response to see if it addresses your concerns? Your timely feedback will be extremely valuable to us. Could you read and let us know if there are more questions? We would be very grateful! Your decision is of utmost importance to us, and we earnestly hope that you will consider the significant contribution of this research to the field of image compression. Thank you very much!
> >
> > Best regards,
> >
> > All the authors

---

> > > ### Comment · Reviewer_qmzM · 2024-11-26
> > >
> > > Overall, I believe this is a strong paper that introduces an innovative approach to adjusting bitrate within a unified framework using mask-ratio strategies. The authors' response has addressed most of my concerns, and I look forward to seeing the final manuscript incorporate all the experimental results.

---

> > > > ### Author Response · Authors · 2024-11-28
> > > > **Response to Reviewer qmzM**
> > > >
> > > > We are very grateful for your approval of our work. Thank you for providing the code of the [Ref1]. We have detailed the comparison with [Ref1] as follows:
> > > >
> > > > | Dataset | | Kodak | | CLIC2020 |
> > > > |:-----------:|:-------:|:-------:|:-------:|:-------:|
> > > > | Method | Ours | Mao *et al.* [Ref1] | Ours | Mao *et al.* [Ref1]
> > > > | Bpp | 0.0381 | 0.0391| 0.0372 | 0.0389 |
> > > > | LPIPS | **0.115** | 0.136 | **0.086** | 0.112 |
> > > >
> > > > The above quantitative results are reported in **Table 3 (Appendix A.6, $\textcolor[RGB]{184,134,11}{highlighted\ in\ gold}$)** in the revised manuscript. Our method achieves the best LPIPS on both datasets with lower bpp, validating its superiority. We have also updated the above results with qualitative comparisons in our revised manuscript **(Appendix A.6, Figure 11, $\textcolor[RGB]{184,134,11}{highlighted\ in\ gold}$)** and made sure to keep them in the final version.
> > > >
> > > > Besides, we apologize for the error of [Ref5], which is revised in our manuscript.
> > > >
> > > > We believe that your and all reviewers' comments will make our work even better. Thanks again.
> > > >
> > > > [Ref1] Qi Mao, *et al.*, Extreme image compression using fine-tuned vqgan models. DCC, 2024.

---

> ### Author Response · Authors · 2024-11-21
> **Response to Reviewer qmzM (Part 2)**
>
> **Weakness 2**: More comparisons in Figure 4.
>
> **Response 2**: We apologize that some results are missing. We have included the additional comparisons in **Figure 4 (revised manuscript, $\textcolor[RGB]{65, 105, 225}{highlighted\ in\ blue}$)**. Our Control-GIC achieves comparable performance in most metrics and exhibits a significant improvement in NIQE. Notably, compared to SCR and CTC, Control-GIC achieves finer granular flexibility for bitrate control while preserving obvious preferable perceptual quality. Besides, since conventional CNN-based NIC methods, *e.g.* M&S (Hyperprior) and SCR, optimize for the R-D trade-off using pixel-wise MSE loss, one can see that they produce relatively higher PSNR than HiFiC and CDC as well as ours.
>
> We also include more generative models (MRIC [Ref1] and MS-ILLM [Ref2]) in Figure 4. These two methods are trained for specific R-D points, which achieve relatively higher quantitative performance in LPIPS and DISTS. However, even though, our method exhibits comparable performance in FID and KID, and better NIQE and model efficiency than these two most state-of-the-art methods. Here, for a simple clarity, we provide training steps (M), BD-rate (BD-LPIPS) results on the DIV2K dataset and encoding/decoding time (s) on the Kodak dataset for quick comparison using VVC as an anchor, as follows:
> | Method | Training Steps (M) | BD-rate (BD-LPIPS)↓ | Encoding time (s) | Decoding time (s) |
> |:-------:|:-------:|:-------:|:-------:|:-------:|
> | BPG | - | 34.5704 | 1.1001 | 0.3514 |
> | M&S | 2n | 33.4252 | 0.0923 | 0.0514 |
> | SCR | 9.4 | 38.9885 | 0.7924 | 0.6477 |
> | CTC | 3 | -7.8118 | 3.5067 | 2.0880 |
> | HiFiC | 2n | -72.6776 | 0.5306 | 1.4993 |
> | CDC | 1.5n |-48.2543 | 0.1069 | 1.1054 |
> | MRIC | 3n |-81.9656 | 0.0598 | 0.0352 |
> | MS-ILLM | 2n |-74.0735 | 0.1094 | 0.0713 |
> | Ours | **0.6** | -67.5585 | **0.0143** | **0.0244** |
>
> It can be seen that the variable-rate approach represented by SCR does not perform as well as models trained on a single point (*e.g.*, M&S), while CTC obtains superior performance with a larger number of parameters (399M, computed from the official open-source code), but with a very large drop in model efficiency (the encoding time is $38\times$ times longer and the decoding time is $40\times$ times longer than M&S). MRIC and MS-ILLM trained separately for each compression rate obtain very close BD-rate savings with our **Control-GIC**, outperforming other methods. Our method achieves the fastest encoding/decoding time, which is $7\times$ faster than MS-ILLM and $4\times$ faster than MRIC in encoding, and $3\times$ faster than MS-ILLM and $1.5\times$ faster than MRIC in decoding. Moreover, the single-point training methods (*e.g.*, HiFiC, CDC, MRIC, MS-ILLM) require independent training of $n$ models for $n$ R-D points. Our proposed model requires only a single training session that enables compression across various bitrates, with the total training steps being substantially reduced to 0.6 million steps. By comparison, our method can achieve a promising balance among training costs, inference speed, and BD-rate saving.
>
> **[Ref1]** Eirikur Agustsson *et al.*, Multi-realism image compression with a conditional generator, CVPR, 2023.
>
> **[Ref2]** Matthew Muckley *et al.*, Improving Statistical Fidelity for Neural Image Compression with Implicit Local Likelihood Models, ICML, 2023.

---

> ### Author Response · Authors · 2024-11-21
> **Response to Reviewer qmzM (Part 3)**
>
> **Weakness 3**: Comparison with VQGAN methods
>
> **Response 3**: Thanks for your valuable suggestion. The main reason is that our **Control-GIC** and these methods are different from the bpp ranges and bitrate flexibility.
>
> **1) Different bpp ranges**. **[Ref3] and [Ref4]** address the challenge of extremely low bitrate compression with tailored strategies. The encoder in **[Ref3]** employs 16x downsampling and clusters a 16384-sized codebook into {2048, 1024, 512, 256, 128, 64, 32, 16, 8}, managing bitrate by adjusting codebook sizes. Similarly, the encoder in **[Ref4]** uses 16x downsampling, discards about 40% of features, and clusters the codebook of size 16384 into {16, 64, 256}, controlling bitrate through codebook clustering as well. **To clarify the bitrate calculations**, we assume an input image resolution of $256\times256$ pixels, with a uniform distribution of codes within the codebook, meaning all indexes have equal bitstream lengths. **For [Ref3]**, the encoder yields $16\times16$ feature maps with codebook sizes varying from 8 to 2048 ($2^3$ to $2^{11}$), translating to the bit transmission range is [$16\times16\times3$, $16\times16\times11$], and the bpp interval of [0.011, 0.043]. **For [Ref4]**, with a $16\times16$ feature map retaining about 60% of feature points and codebook sizes from 16 to 256 ($2^4$ to $2^8$), the bit transmission range is [$16\times16\times0.6\times4$, $16\times16\times0.6\times8$], resulting in the bpp interval of [0.009, 0.019]. **For [Ref5]**, the upper limit of the bitrate reported in its paper does not exceed 0.035.
>
> Differently, **our Control-GIC** employs a multi-grained encoder with variable downsampling factors—$4\times$ (fine), $8\times$ (medium), and $16\times$ (coarse)—and a fixed codebook size of 1024. We manage the bitrate by adjusting the ratio of these granularities, offering a significant advantage over the aforementioned methods. The lowest bitrate of our method corresponds to a fully coarse-grained partition, yielding a $16\times16$ feature map, while the highest bitrate corresponds to a fully fine-grained partition, resulting in a $64\times64$ feature map. With a codebook size of 1024 ($2^{10}$), the range of bits required for transmission is [$16\times16\times10$, $64\times64\times10$], and the bpp interval is [0.040, 0.625], catering to a broad range of compression requirements and ensuring an optimal balance between compression efficiency and image quality.
>
> **2) Bitrate flexibility**. These methods train the models separately for specific R-D points with Lagrange multiplier $\lambda$-weighted R-D loss, each corresponding to an individual $\lambda$. In this way, multiple fixed-rate models are necessitated to vary bitrates, leading to dramatic computational costs and inefficient deployment to cater to diverse bitrates and devices. In contrast, our method is capable of fine-grained bitrate adaption while preserving high-perceptual fidelity reconstruction in a unified model.
>
> **To investigate the performance of our method, we have conducted experiments on extremely for extremely low bitrate compression (<0.05bpp)**. Since the source codes and pre-trained checkpoints of [Ref3]-[Ref5] are unavailable, we report the best-approximated results of **[Ref3]** on Kodak and CLIC2020 datasets based on the public paper, where quantitative results are as follows:
> | Dataset | | Kodak | | CLIC2020 |
> |:-----------:|:-------:|:-------:|:-------:|:-------:|
> | Method | Ours | Mao *et al.* [Ref3] | Ours | Mao *et al.* [Ref3]
> | Bpp | 0.038 | < 0.04  | 0.038 | < 0.04 |
> | LPIPS | **0.119** | > 0.16 | **0.091** | > 0.13 |
>
> Our method achieves the best LPIPS on both datasets with lower bpp, validating its superiority. Moreover, we also provide several visualizations of compressed images in **Figure 11 (Appendix, $\textcolor[RGB]{184,134,11}{highlighted\ in\ gold}$)**, where the results demonstrate that our method can be well-generalized to low bitrates and maintain vivid reconstruction.
>
> **[Ref3]** Qi Mao, *et al.*, Extreme image compression using fine-tuned vqgan models. DCC, 2024.
>
> **[Ref4]** Naifu Xue, Qi Mao, *et al.*, Unifying Generation and Compression: Ultra-low bitrate Image Coding Via Multi-Stage Transformer. ICME, 2024.
>
> **[Ref5]** Zhaoyang Jia *et al.*, Generative Latent Coding for Ultra-Low Bitrate Image Compression. CVPR, 2024.

---

> > ### Comment · Reviewer_qmzM · 2024-11-26
> >
> > The open-source codes of [Ref3] can be find here:https://github.com/CUC-MIPG/VQGAN-Compression, could you give a more detailed RD comparisons in your final manuscript?

---

> > ### Comment · Reviewer_qmzM · 2024-11-26
> >
> > [Ref5] Zhaoyang Jia et al., is CVPR2024 not WACV2024

---

> ### Author Response · Authors · 2024-11-21
> **Response to Reviewer qmzM (Part 4)**
>
> **Weakness 4**: Results on CLIC2020
>
> **Response 4**: Thanks for your suggestion. We have provided the comparison on the CLIC2020 dataset, where the R-D performance and qualitative performance are shown in **Figure 10 (Appendix, $\textcolor[RGB]{65, 105, 225}{highlighted\ in\ blue}$)**. Here, we provide the results of several representative methods for comparison.:
> | Method | Bpp↓ | LPIPS↓ | DISTS↓ | FID↓ | KID↓ | PSNR↑ | NIQE↓|
> |:-----------:|:-------:|:-------:|:-------:|:-------:|:-------:|:-------:|:-------:|
> | M&S | 0.2921 | 0.1144 | 0.1270 | 33.2748| 0.0164 | 35.0479 | 5.0712 |
> | CTC | 0.2714 | 0.0961 | 0.1100 | 24.4000| 0.0127 | 35.6997 | 4.6345 |
> | HiFiC | 0.2244 | 0.0331 | 0.0532 | 3.4804 | 0.0007 | 31.5913 | 3.8988 |
> | CDC | 0.2284 | 0.0548 | 0.0411| 2.8103 | 0.0006 | 28.7806 | 4.4357 |
> | Ours | 0.2243 | 0.0342 | 0.0503| 3.3644 | 0.0007 | 28.9413 | 3.4638 |
>
> It can be seen that **Control-GIC** achieves superior performance in most metrics over M&S and variable-rate method CTC. Compared to generative compression methods which are trained separately for multiple R-D points, our **Control-GIC** still maintains competitive performance, which validates that our method can achieve optimal trade-off between flexibility and effectiveness.
>
> **Questions (1)**: Justification on the mask ratio setting.
>
> **Response (1)**: In our method, when the granularities of image patches changes, the characteristics of input data processed by the model also change. In actual, for each specific granularity ratio combination of data, the model learns as a new task. Therefore, the model tends to suffer from the **catastrophic forgetting** among multiple combinations, which is a common challenge in deep learning tasks. Regard to our controllable variable bitrate approach, it is necessary to ensure the model for peak performance given a granularity ratio while mitigating the performance degradation for other ratios.
>
> During experimental process, we have researched various combinations for training the model. However, many of them cannot result in even acceptable reconstruction. Then, we rethink the optimization for our model as a geometric optimization problem. Imagine multiple points on a line segment; the point that minimizes the total distance to all other points is the midpoint. Similarly, in our context, we aim to position our model on the R-D curve such that its parameters are as close as possible to those of a model trained at any given bitrate point. The optimal point for this is the midpoint, which serves as a balanced and versatile reference point for model performance across different bitrates. As a result, we get the ratio setting (fine: 50%, medium : 40%, and coarse: 10%), which can yield optimal results.
>
> **Questions (2)**: Explanation for the bpp range.
>
> **Response (2)**: Similar to the explanation in **Response 3**, where the lowest bitrate ($<0.05$ bpp) of our method corresponds to a fully coarse-grained partition, *i.e.* $(r_1, r_2, r_3)$ = (0, 0, 100%) for fine, medium, and coarse. We use Mao *et al.* **[Ref3]**, a VQGAN-based method also for very low bitrate, as a reference. The quantitative results can also be seen in Response 3.
>
> **[Ref3]** Qi Mao, *et al.*, Extreme image compression using fine-tuned vqgan models. DCC, 2024.

---

### Official Review · Reviewer_o1HZ · 2024-11-05

**Soundness:** 3
**Presentation:** 3
**Contribution:** 2
**Rating:** 6
**Confidence:** 5

**Summary:**

The paper titled "ONCE-FOR-ALL: CONTROLLABLE GENERATIVE IMAGE COMPRESSION WITH DYNAMIC GRANULARITY ADAPTION" introduces Control-GIC, a framework for controllable generative image compression. It addresses the challenge of flexible rate adaption in image compression by leveraging a VQGAN framework that encodes images as variable-length codes . The framework correlates local image patch information density with granular representations, allowing for fine-grained bitrate control. It includes a granularity-informed encoder, a statistical entropy coding module, and a probabilistic conditional decoder. The experiments demonstrate that Control-GIC outperforms state-of-the-art methods in terms of flexibility, perceptual quality, and compression efficiency.

**Strengths:**

1. Control-GIC combines classical coding principles with VQGAN to achieve controllable generative compression across various bitrates with a unified model.
2. The framework allows for highly flexible and controllable bitrate adaption, which is a significant advancement over existing methods.
3. Unlike other methods that require training multiple models for different bitrates, Control-GIC can adapt to various bitrates with a single model, reducing computational costs.

**Weaknesses:**

1. More comparion with other GIC methods need to be provided.
2. The novelty is limited compared to other VQGan based GIC method

**Questions:**

See weakness.

---

> ### Author Response · Authors · 2024-11-21
> **Response to Reviewer o1HZ (Part 1)**
>
> **Weakness 1**: More comparisons with other GIC methods.
>
> **Response 1**: We have compared our method with two additional latest GIC methods, MRIC [Ref1] and MS-ILLM [Ref2] on both Kodak and DIV2K datasets, **where the results are shown in **Figure 3 - 5** (revised manuscript, $\textcolor[RGB]{65, 105, 225}{highlighted\ in\ blue}$)**, respectively. One can see that, these two methods are trained for specific R-D points, which achieve relatively higher quantitative performance in LPIPS and DISTS. However, even though, our method exhibits comparable performance in FID and KID, and better NIQE and model efficiency than these two most state-of-the-art methods. Here, for simple clarity, we provide training steps (M), BD-rate (BD-LPIPS) results on the DIV2K dataset, and encoding/decoding time (s) on the Kodak dataset for quick comparison using VVC as an anchor, as follows:
> | Method | Training Steps (M) | BD-rate (BD-LPIPS)↓ | Encoding time (s) | Decoding time (s) |
> |:-------:|:-------:|:-------:|:-------:|:-------:|
> | BPG | - | 34.5704 | 1.1001 | 0.3514 |
> | M&S | 2n | 33.4252 | 0.0923 | 0.0514 |
> | SCR | 9.4 | 38.9885 | 0.7924 | 0.6477 |
> | CTC | 3 | -7.8118 | 3.5067 | 2.0880 |
> | HiFiC | 2n | -72.6776 | 0.5306 | 1.4993 |
> | CDC | 1.5n |-48.2543 | 0.1069 | 1.1054 |
> | MRIC | 3n |-81.9656 | 0.0598 | 0.0352 |
> | MS-ILLM | 2n |-74.0735 | 0.1094 | 0.0713 |
> | Ours | **0.6** | -67.5585 | **0.0143** | **0.0244** |
>
> It can be seen that the variable-rate approach represented by SCR does not perform as well as models trained on a single point (*e.g.*, M&S), while CTC obtains superior performance with a larger number of parameters (399M, computed from the official open-source code), but with a very large drop in model efficiency (the encoding time is $245\times$ times longer and the decoding time is $85\times$ times longer than ours). MRIC and MS-ILLM trained separately for each compression rate obtained very close BD-rate savings with our **Control-GIC**, outperforming other methods. Our method achieves the fastest encoding/decoding time, which is $7\times$ faster than MS-ILLM and $4\times$ faster than MRIC in encoding, and $3\times$ faster than MS-ILLM and $1.5\times$ faster than MRIC in decoding. Moreover, the single-point training methods (*e.g.*, HiFiC, CDC, MRIC, MS-ILLM) require independent training of $n$ models for $n$ R-D points. Our proposed model requires only a single training session that enables compression across various bitrates, with the total training steps being substantially reduced to 0.6 million steps. By comparison, our method can achieve a promising balance among training costs, inference speed, and BD-rate saving.
>
> **[Ref1]** Eirikur Agustsson *et al.*, Multi-realism image compression with a conditional generator, CVPR, 2023.
>
> **[Ref2]** Matthew Muckley *et al.*, Improving Statistical Fidelity for Neural Image Compression with Implicit Local Likelihood Models, ICML, 2023.

---

> ### Author Response · Authors · 2024-11-21
> **Response to Reviewer o1HZ (Part 2)**
>
> **Weakness 2**: Discussion with VQGAN-based methods.
>
> **Response 2**: Here, we would like to discuss with existing VQGAN-based compression methods [Ref3] - [Ref5], to clarify the differences and our contributions.
>
> **First**, Mao *et al.* [Ref3] and UIGC [Ref4] control the bitrate by **adjusting the codebook size**, which necessitates the fine-tuning on the model for each codebook size to achieve variable bitrates. GLC [Ref5] introduces VQVAE in the hyperprior model to mine the semantic visual components and guarantee the reconstruction fidelity, which also requires multiple trainings with different $\lambda$ to realize multiple bitrates. In contrast, our method **only requires training once and does not involve any fine-tuning**, which enables to produce the compression results across a broad spectrum of bitrates.
>
> **Second**, these methods are motivated by the representative and generative ability of VQGAN, which ignores explicitly exploring the local image characteristics and the differences between patches (*e.g.*, local information density). Our method **correlates the information density of local image patches with their granular representations**. Hence, we can flexibly determine a proper allocation of granularity for the patches to achieve **dynamic adjustment for VQ-indices**, resulting in desirable compression rates. Our method assigns different granularities to regions of different complexity and achieves fine control of the bitrate through the dynamic adjustment of granularities.
>
> **Third**, to our knowledge, the mentioned knowledge above focuses on extremely low bitrate compression (<0.05bpp), whereas our method is prone to support a more general bit rate range within 0.1~0.6bpp. Besides, to investigate the effectiveness of our method at such a low bitrate. **In Appendix A.6**, we set the granularity of all patches as coarse ones, *i.e.*, the granularity for fine, medium, and coarse are $(r_1, r_2, r_3)$ = (0, 0, 100%), resulting in the theoretical lowest bitrate. We report the quantitative results compared to **[Ref3]** in Bpp and LPIPS on the Kodak and CLIC2020 datasets, which are shown in **Table 3 (Appendix A.6,   $\textcolor[RGB]{184, 134, 11}{highlighted\ in\ gold}$)**.
> | Dataset | | Kodak | | CLIC2020 |
> |:-----------:|:-------:|:-------:|:-------:|:-------:|
> | Method | Ours | Mao *et al.* [Ref3] | Ours | Mao *et al.* [Ref3]
> | Bpp | 0.0381 | 0.0391| 0.0372 | 0.0389 |
> | LPIPS | **0.115** | 0.136 | **0.086** | 0.112 |
>
> One can see that our method achieves the best LPIPS on both datasets with lower bpp, validating its superiority. Moreover, we also provide several visualizations of compressed images in **Figure 11 (Appendix A.6, $\textcolor[RGB]{184,134,11}{highlighted\ in\ gold}$)**, where the results demonstrate that our method can be well-generalized to low bitrates and maintain vivid reconstruction.
>
> **[Ref3]** Qi Mao, *et al.*, Extreme image compression using fine-tuned vqgan models. DCC, 2024.
>
> **[Ref4]** Naifu Xue, Qi Mao, *et al.*, Unifying Generation and Compression: Ultra-low bitrate Image Coding Via Multi-Stage Transformer. ICME, 2024.
>
> **[Ref5]** Zhaoyang Jia, *et al.*, Generative Latent Coding for Ultra-Low Bitrate Image Compression. CVPR, 2024.

---

> ### Author Response · Authors · 2024-11-24
> **Request for your feedback**
>
> Dear Reviewer o1HZ:
>
> Hoping this message finds you well. Your comments and the provided insights are very valuable to our work. Besides previous responses and revisions, we now add more experiments and analyses, including 1) well-developed theoretical support and detailed descriptions of the granularity assignments, and 2) results from additional datasets. We will attach these additional experiments and analyses to our final version. As the rebuttal-discussion period is nearing its end, could you please review our response to see if it addresses your concerns? Your timely feedback will be extremely valuable to us. Could you read and let us know if there are more questions? We would be very grateful! Your decision is of utmost importance to us, and we earnestly hope that you will consider the significant contribution of this research to the field of image compression. Thank you very much!
>
> Best regards,
>
> All the authors

---

### Meta-Review · Area_Chair_zJ6B · 2024-12-16

**Metareview:**

The paper received mixed reviews from five experts.

The authors' provided responses trying to address the reviewers' concerns.

Particularly, Reviewer P96B decided to lower significantly her/his ratings after receiving the authors' responses without giving any explanation to the authors/the fellow reviewers.
Therefore, Reviewer P96B's ratings are to be ignored.

The other reviewers are satisfied with the responses, either keeping or elevating their initial ratings.

After rebuttal, Reviewer qmzM champions acceptance (8: accept, good paper) while Reviewers o1HZ, WUkA, and bPMe are borderline, generally appreciating the contributions and not having remaining important concerns.

AC finds merits to the paper (Reviewer qmzM : "a strong paper that introduces an innovative approach to adjusting bitrate within a unified framework using mask-ratio strategies") and recommends acceptance and invites the authors to refine the paper contents so that most of the concerns are addressed in the paper.

**Additional Comments On Reviewer Discussion:**

Reviewer P96B decided to lower significantly her/his ratings after receiving the authors' responses without giving any explanation to the authors/the fellow reviewers.
Therefore, Reviewer P96B's ratings are to be ignored.

The other reviewers appreciated the authors' responses and extra information, most of the reviewers' concerns being properly addressed by the authors.

---

### Decision · Program_Chairs · 2025-01-22

Accept (Poster)